# A Hypertoroidal Covering for Perfect Color Equivariance

**Yulong Yang** [* 1]   **Zhikun Xu** [* 1 2]   **Yaojun Li** [1]   **Christine Allen-Blanchette** [1]

## Abstract

When the color distribution of input images changes at inference, the performance of conventional neural network architectures drops considerably. A few researchers have begun to incorporate prior knowledge of color geometry in neural network design. These color equivariant architectures have modeled hue variation with 2D rotations, and saturation and luminance transformations as 1D translations. While this approach improves neural network robustness to color variations in a number of contexts, we find that approximating saturation and luminance (interval valued quantities) as 1D translations introduces appreciable artifacts. In this paper, we introduce a color equivariant architecture that is truly equivariant. Instead of approximating the interval with the real line, we lift values on the interval to values on the circle (a double-cover) and build equivariant representations there. Our approach resolves the approximation artifacts of previous methods, improves interpretability and generalizability, and achieves better predictive performance than conventional and equivariant baselines on tasks such as fine-grained classification and medical imaging tasks. Going beyond the context of color, we show that our proposed lifting can also extend to geometric transformations such as scale.

## 1. Introduction

While shape and geometry dominate humans object perception, color information plays an increasing role for fine grained visual tasks (Bramão et al., 2011). However, until recently (Lengyel et al., 2021; 2023; Yang et al., 2024), principled design based on perceptual information has attracted less attention when designing networks even though its empirical importance for performance was noted early

on (Engilberge et al., 2017).

A popular heuristic for dealing with perceptual variations is to simply transform the images to gray scale (Güneş et al., 2016; Xie & Richmond, 2018). However, the assumption that training and testing images are drawn from the same distribution limits the generalizability of these models. Data augmentation techniques (Hendrycks et al., 2021) seek to find a systematic way to expand the training dataset to capture more variations, but at the expense of increased training time and computation resources. Color invariant networks (Lengyel et al., 2021) leverage the geometric structure of color to ensure that perceptual variations do not affect network outputs. This however discards important information that is especially valuable on tasks like fine-grained classification. Color equivariant approaches allow the network to retain color information by leveraging the geometric structure of color. However, existing techniques are only able to be approximately equivariant to only hue (Lengyel et al., 2023) or exhibits artifacts when dealing with shifts in saturation and luminance (Yang et al., 2024). These drawbacks result from the fact that, unlike hue, saturation and luminance groups do not exhibit cyclic behavior.

In this paper, we propose hypertorodoidal color equivariant network ($\mathbb{T}^3$CEN) that is perfectly equivariant to shifts in hue, saturation, and luminance by leveraging the geometric structure of the Hue-Saturation-Luminance (HSL) color space. We leverage topological covering to reformulate the saturation and luminance group such that they exhibit cyclic behavior that enables perfect equivariance. The improved structure of the network yields a more interpretable latent space which results in improved performance in image classification and segmentation tasks. We showcase the improved performance on both synthetic datasets with variation in one channel as well as large scale real world medical datasets, where $\mathbb{T}^3$CEN outperforms baseline equivariant and conventional architectures. Finally, we also show that the proposed double-cover can be applied to achieve perfect equivariance to geometric transformations such as scale.

## 2. Related Works

**Group convolutional neural networks.** The widespread adoption of convolutional networks for image processing task can partially be attributed to the improved generaliza-

---

*Equal contribution [1]Princeton University [2]Tsinghua University.. Correspondence to: Y. Yang <yulong.yang@princeton.edu>.

*Proceedings of the 43$^{rd}$ International Conference on Machine Learning*, Seoul, South Korea. PMLR 306, 2026. Copyright 2026 by the author(s).

tion from planar translational equivariance. Many existing architectures have focuses on expanding the equivariance capacity of CNNs to additional symmetry groups (Kondor & Trivedi, 2018; Cohen et al., 2019b) such as rotation and scale. In their seminal work, Cohen & Welling (2016) introduced an architecture for finite group convolutions for 2D rotations and reflections by convolving inputs with the group orbit of the learned filter bank. Worrall et al. (2017) further introduced equivariance to continuous rotations by leveraging circular harmonic structure.

Group convolutional networks have also been designed to accommodate non-cyclic group such as scale symmetry (Esteves et al., 2018; Worrall & Welling, 2019; Sun & Blu, 2023) and groups acting on higher dimensions such as 3D rotations (Thomas et al., 2018; Esteves et al., 2019), Euclidean transformations (Batzner et al., 2022), and general manifolds (Cohen et al., 2019a). Leveraging symmetries that occur naturally in computer vision tasks, these group convolutional networks offer improved interpretability, generalizability, and data efficiency over conventional convolutional networks.

**Learning under color imbalance.** Existing studies have demonstrated the negative impact of color variance on classification tasks (Buhrmester et al., 2019; De & Pedersen, 2021). Previous works attempt to address this challenge through data augmentation algorithms, such as AugMix (Hendrycks et al., 2019) and Deep Augment (Hendrycks et al., 2021), to artificially expand the training data. More structured and data efficient approaches leverage invariant architectures to improve robustness and generalization to changes in color and other perceptual quantities (Geusebroek et al., 2001; Chong et al., 2008; Lengyel et al., 2021; Pakzad et al., 2022; Hong et al., 2024). However, retaining color information can add an important cue in representation learning (Engilberge et al., 2017), which is in conflict with the main aim of color invariance.

Equivariant architectures resolve this conflict by retaining color information throughout the network. Lengyel et al. (2023) leverages this concept to propose convolutional layers that are equivariant to hue-shifts by rotating the RGB values of input images in the lifting layer. However, performing lifting directly on the input color space yields representations that are only approximately equivariant to only hue-shifts. Yang et al. (2024) resolves this drawback by performing lifting operations on the HSL color space to design a convolution layer that is equivariant to hue-, saturation-, and luminance-shifts. Hue is modeled with a cyclic group, and saturation and luminance with a translation group, with learned representations that are exactly equivariant to hue-shift and only approximately so to saturation-and luminance-shifts.

Our proposed hypertoroidal color equivariant network dif-

fers from Yang et al. (2024) in two main aspects. First, we propose a new topological covering map that gives saturation and luminance groups cyclic characteristics; and second we use this covering to design a convolution network that is fully equivariant to shifts in hue, saturation, and luminance.

**Topological covering.** A covering is a map between topological spaces that intuitively projects multiple copies onto itself. A notable application of cover map is the double-cover from unit quaternions ($\mathbb{S}^3$) to 3D rotations ($SO(3)$) (Gallier & Quaintance, 2019). Covering maps can be leveraged to project a complex space to one with more desirable characteristics. Specifically, machine learning applications have leveraged covering maps to design well behaved convolutions on geometric surfaces (Maron et al., 2017; Haim et al., 2019) and characterize graph neural network expressivity (Gallier & Quaintance, 2019; Garg et al., 2020). In this paper, we propose a convolutional network that is perfectly equivariant to color by leveraging a covering that enables perfect group convolutions for non-cyclic channels.

## 3. Background

This section reviews key concepts including groups, group actions, equivariance, group convolution, and the notion of a topological covering.

**Groups and group actions.** Group convolutional neural networks (GCNNs) (Cohen & Welling, 2016) demonstrate how the conventional planar CNN can be generalized to finite groups. Here, a *group* is defined as a set $G$ together with a binary operation $\cdot : G \times G \to G$ that satisfies the following axioms:

1. *Associativity.* For $a, b, c \in G$, $(a \cdot b) \cdot c = a \cdot (b \cdot c)$,

2. *Existence of identity.* There exists an element $e \in G$, so that for any element $a \in G$, $e \cdot a = a \cdot e = a$,

3. *Existence of inverse.* For any element $a \in G$ there exists an element $b \in G$ so that $a \cdot b = b \cdot a = e$.

The elements of a group transform elements of a (left) $G$-set $X$, by the *group action* $\varphi : G \times X \to X$ which satisfies the following:

1. For all $a, b \in G$ and all $x \in X$, $\varphi(a, \varphi(b, x)) = \varphi(ab, x)$

2. For all $x \in X$, $\varphi(e, x) = x$.

**Equivariance.** An important characteristic of conventional planar CNNs is that they are equivariant to translation. Here, *equivariance* is defined as a relationship between two $G$-sets $X$ and $Y$. Given the group actions $\varphi : G \times X \to X$ and $\phi : G \times Y \to Y$, a function $f : X \to Y$ is said to be equivariant, if and only if for all $x \in X$, and $a \in G$, $f(\varphi(a, x)) = \phi(a, f(x))$. Invariance is a special case of equivariance where $\phi(a, f(x)) = f(x)$.

**Group convolution.** In a conventional CNN, the feature map at layer $l$, denoted $f^l : \mathbb{Z}^2 \to \mathbb{R}^{K^l}$, is convolved with a set of $K^{l+1}$ filters, denoted $\psi_i^l : \mathbb{Z}^2 \to \mathbb{R}^{K^l}$, where $i$ ranges from 1 to $K^{l+1}$. The feature map $f^{l+1}$ is the result of convolving feature map $f^l$ with filters $\psi_i^l$,

$$[f^l * \psi_i^l](x) = \sum_{y \in \mathbb{Z}^2} \sum_{k=1}^{K^l} f_k^l(y) \psi_{i,k}^l(x - y), \ x \in \mathbb{Z}^2. \quad (1)$$

This operation can be shown to be equivariant to 2D translations. The more general group convolution,

$$[f^l * \psi_i^l](a) = \sum_{b \in G} \sum_{k=1}^{K^l} f_k^l(b) \psi_{i,k}^l(b^{-1}a), \ a \in G, \quad (2)$$

can be shown to be equivariant to the action of the group $G$.

**Topological covering.** GCNNs are a powerful tool, but they require that symmetry transformation is a finite group. There are several contexts in which this requirement does not hold (e.g., color symmetry and scale symmetry). In cases where the symmetry is interval valued, we can construct a covering that can be used to lift the interval which is not a group to the circle which is a group. A map is a *covering map* iff it is smooth and surjective, and satisfies criteria for evenness (see (Gallier & Quaintance, 2020) for a precise definition).

## 4. Background

In this section we introduce our hypertoroidal color equivariant network ($\mathbb{T}^3$CEN). We present the definitions for the hue, saturation and luminance groups, and their group actions. Next we review HSL group convolution, and finally, we introduce our lifting layer and prove it is equivariant.

**Hue group and group action.** Following Yang et al. (2024), we identify elements of the discretized hue group $H_N$ with elements of the cyclic group $C_N$. Given an HSL image $x = (x_h, x_s, x_l)$ where $x_h$, $x_s$, and $x_l$ are the hue, saturation and luminance channels, we define the action of $h_i \in H_N$ on $x$ as,

$$\varphi_h(h_i, x) = ((x_h + h_i) \,(\text{mod } 255), x_s, x_l), \quad (3)$$

where $(\cdot)\text{mod}(\cdot)$ denotes pixel-wise modulus. Given a function $f = (f_1, \ldots, f_N)$ on $H_N$ (e.g., a feature map, or a filter), we define the action of $h_i$ on $f$ as,

$$\phi_h(h_i, f) = \left(f_{(1+i)(\text{mod } N)}, \ldots, f_{(N+i)(\text{mod } N)}\right). \quad (4)$$

**Saturation group and group action.** In the HSL color space, saturation values are restricted to an interval. Because the interval does not have group structure, it is not possible to construct an GCNN on this representation directly. To get around this, (Yang et al., 2024) model saturation with

the structure of the translation group $(\mathbb{R}, +)$. An element $s_i$ of resulting saturation group $S_M$, acts on HSL images by the group action

$$\varphi_s(s_i, x) = (x_h, \min(x_s + s_i, c), x_l), \quad (5)$$

and on $S_M$ functions $f = (f_1, \ldots, f_M)$ by the group action

$$\phi_s(s_i, f) = (f_{1+i}, \ldots, f_M, \underbrace{\mathbf{0}, \ldots, \mathbf{0}}_{i}). \quad (6)$$

However, because saturation values are bounded, enforcing the translation group requires value clipping (see equation (5) and (6)). Consequently, saturation transformations can introduce spurious artifacts making learned representations only approximately equivariant.

To remedy this, we model saturation with the structure of the cyclic group $C_M$. Given the interval of valid saturation values $I = [0, c]$, we first center the interval giving a new interval $\tilde{I} = I - c/2$, then we construct the saturation manifold $\tilde{S}$ using the inverse of the double-cover $\pi : \mathbb{T}^1 \to \tilde{I}$, where $\pi(\theta) = \frac{c}{2} \sin \theta$. From there, we define the saturation group $(S_M, \cdot)$, where the set $S_M$ of order $M$ is determined by a uniform discretization of $\tilde{S}$, and the binary operation $\cdot : S_M \times S_M \to S_M$ is defined,

$$a \cdot b \mapsto (a + b) \bmod 2\pi, \quad a, b \in S_M. \quad (7)$$

We also define the $S_M$ group action on HSL images and functions on $S_M$. Given an HSL image $x = (x_h, x_s, x_l)$, we define the action of $s_i \in S_M$ on $x$ by,

$$\varphi_s(s_i, x) = (x_h, \pi((x_s + s_i)(\bmod 2\pi)), x_l), \quad (8)$$

and given a function $f = (f_1, ..., f_M)$ on the discrete saturation group $S_M$, we define the action of $s_i$ on $f$ by,

$$\phi_s(s_i, f) = \left(f_{(1+i)(\bmod M)}, \ldots, f_{(M+i)(\bmod M)}\right). \quad (9)$$

**Luminance group and group action.** In HSL, luminance values are also restricted to the interval. To give the luminance space group structure, we use the same strategy we used for the saturation space in the previous section. We denote the discretized luminance group $L_R$, the luminance group action on images $\varphi_l(l_i, x)$ and the luminance group action on an $L_R$ function, $\phi_l(l_i, f)$ (see Appendix A.1 for additional details).

**HSL group and group Action.** Having defined the hue, saturation and luminance groups, and their group actions, we can now define the HSL group and its group action. We define the HSL group as the product group $HSL_{NMR} := H_N \times S_M \times L_R$, and the HSL group actions as compositions of the hue, saturation, and luminance group actions. Concretely, given an HSL image $x$, we define the action of $g_{ijk} \in HSL_{NMR}$ on $x$ by

$$\varphi_{hsl}(g_{ijk}, x) = \varphi_h(h_i, \varphi_s(s_j, \varphi_l(l_k, x))), \quad (10)$$

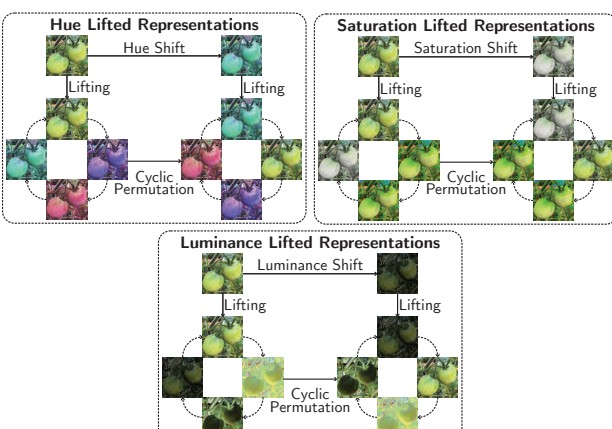

*Figure 1.* **Hue, saturation, and luminance lifting.** We lift an input image with respect to the hue, saturation, and luminance channels. Hue lifting of $\mathbb{T}^3$CEN, which follows the hue lifting proposed in Yang et al. (2024). Saturation and luminance lifting of $\mathbb{T}^3$CEN using a double-cover to give cyclic behavior to the saturation and luminance group. A hue, saturation, or luminance shifted input yields a cyclically permuted lifted representation.

and given an $HSL_{NMR}$ function $f = (f_{111}, \ldots, f_{NMR})$, we define the action of $g_{ijk}$ on $f$ by

$$\phi_{hsl}\left(g_{ijk}, f\right) = \tilde{\phi}_h\left(h_i, \tilde{\phi}_s\left(s_j, \tilde{\phi}_l\left(l_k, f\right)\right)\right), \quad (11)$$

where the modified hue, saturation, and luminance group actions are defined as,

$$\tilde{\phi}_h\left(h_i, f\right) = \left(f_{(1+i)(\text{mod } N)::}, .., f_{(N+i)(\text{mod } N)::}\right), \quad (12)$$

$$\tilde{\phi}_s\left(s_j, f\right) = \left(f_{:(1+j)(\text{mod } M):}, .., f_{:(M+j)(\text{mod } M):}\right), \quad (13)$$

$$\tilde{\phi}_l\left(l_k, f\right) = \left(f_{::(1+k)(\text{mod } R)}, .., f_{::(R+k)(\text{mod } R)}\right). \quad (14)$$

**Lifting layer.** Group convolutional processing is defined between functions on the relevant group. To map input images $x$, to the HSL group we use an HSL lifting layer

$$f^0\left(g_{ijk}\right) = \varphi_{hsl}\left(g_{ijk}, x\right), \quad g_{ijk} \in HSL_{NMR} \quad (15)$$

The resulting feature map $f^0$ is a function on the HSL group and can be used in convolutional processing. Our lifting layer is distinguished from others in the literature in that it can be applied to spaces that have interval structure. Since inputs to the group convolution operator must have group structure, our lifting layer first constructs a double-cover of the interval, to a resulting space that is isomorphic to $\mathbb{T}^1$ (see paragraph Saturation group and group action for additional details). The interval is not a group but $\mathbb{T}^1$ is, and from here we can use a conventional lifting approach to realize a function on the HSL group. Our proposed HSL lifting layer is illustrated in Figure 1.

**HSL group convolution.** Lifted inputs are function on the HSL group and can convolved with other functions

on the HSL group (e.g., HSL filters). Using the definition of group convolution in Equation (2), we define the HSL group convolution as

$$\left[f^l * \psi_i^l\right](a) = \sum_{r \in HSL} \sum_{k=1}^{K^l} f_k^l(r)\, \psi_{i,k}^l\left(r^{-1}a\right). \quad (16)$$

It can be shown that the HSL group convolution is equivariant to hue, saturation and luminance.

# 5. Experiments

In this section, we compare our $\mathbb{T}^3$CEN to various color equivariant and non-equivariant baselines. We show that our model has lower equivariance error than other models on synthetic datasets, and better robustness to color shift on real datasets. We also analyze our double-cover lifting layer, quantifying coverage of the transformation space, and introducing a principled approach for selecting group order based on dataset characteristic. Finally, we demonstrate the utility of our double-cover lifting layer in the contexts of RGB color, and scaling transformations.

## 5.1. Equivariance Error

We first compare the equivariance error of our approach and the state-of-the-art color equivariant method, which we denote as LCER (Yang et al., 2024). We do this both qualitatively and quantitatively. In our qualitative assessment,

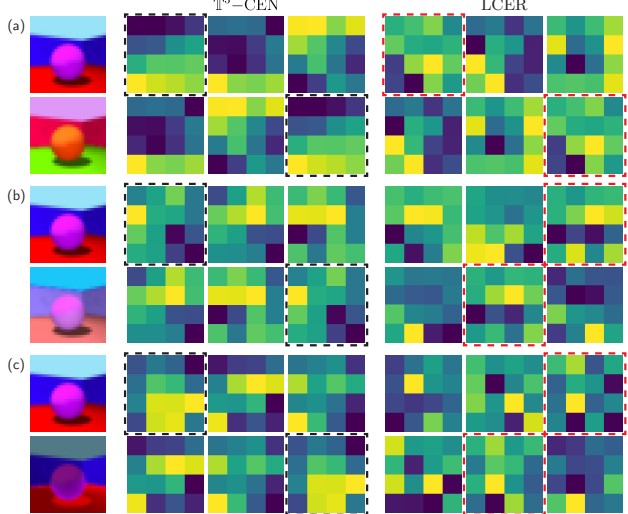

*Figure 2.* **$\mathbb{T}^3$CEN and LCER feature maps under HSL shifts.** The features maps of $\mathbb{T}^3$CEN are equivariant to shifts in hue, saturation, and luminance, while the feature maps of LCER are only equivariant to shifts in hue. **(a)** The images are related by a $90°$ hue rotation. **(b)** The images are related by a $0.5$ shift in saturation. **(c)** The images are related by a $0.5$ shift in luminance. In all cases, because our $\mathbb{T}^3$CEN network is equivariant to color shifts (i.e., hue, saturation, and luminance shifts), the feature maps transform predictably (cyclically permuted). Conversely, LCER is only equivariant to hue shifts.

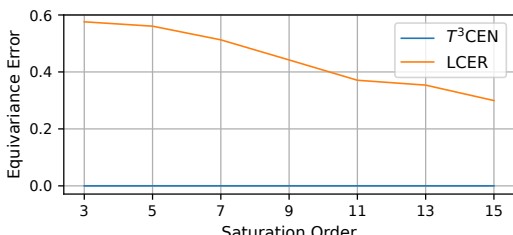

*Figure 3.* **Saturation equivariance error.** The normalized saturation equivariance error for $\mathbb{T}^3$CEN and LCER is reported. $\mathbb{T}^3$CEN has average error $4.66 \times 10^{-6}$ compared to LCER at $0.445$.

we take synthetically generated images from the 3D Shapes dataset (Kim & Mnih, 2018) and transform them in hue, saturation and luminance. If the proposed group actions are color equivariant, we should observe a commutative relationship between transformations of the input and transformations of their respective feature maps. We show the results of this experiment in Figures 2(a)-(c). Our method preserves equivariance for all subgroups of the HSL group, while LCER only preserves equivariance for hue shifts.

In our quantitative assessment, we compute the saturation equivariance error. Following Yang et al. (2024), we define the saturation equivariance error

$$\Delta_{\varphi_s} = \frac{|f(\varphi_s(s_i, x)) - \phi_s(s_i, f(x))|}{|f(\varphi_s(s_i, x)) + \phi_s(s_i, f(x))|}. \quad (17)$$

We plot the saturation equivariance error as a function of the saturation group order $M \in \{3, 5, 7, 9, 11, 15\}$ in Figure 3. The average error of $\mathbb{T}^3$CEN is $4.66 \times 10^{-6}$ and the average error of LCER is $0.445$. We attribute this difference in equivariance error to the lifting error of LCER. We measure the lifting error of the proposed group actions using the mean of the difference $\|x - \phi(g^{-1}, \phi(g, x))\|$. The lifting error of $\mathbb{T}^3$CEN and LCER are compared Figure 4, where

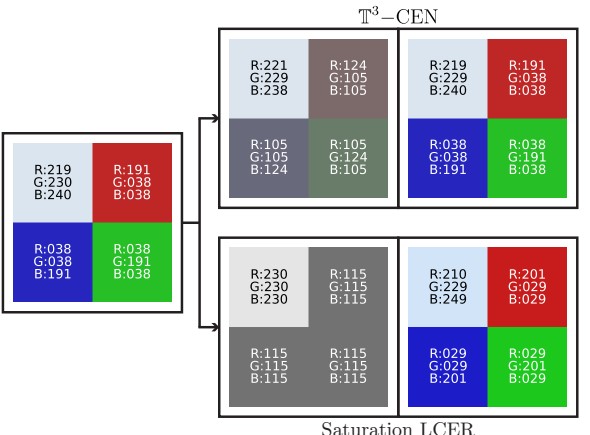

*Figure 4.* **Lifting error comparison.** An input images is lifted to the respective saturation group, shifted down by $0.75$, and shifted up by $0.75$. We compare the restored and original input image for $\mathbb{T}^3$CEN (**top**) and LCER (**bottom**). The average 8-bit integer RGB error is $6.33 \times 10^{-6}$ for $\mathbb{T}^3$CEN and $8.65$ for LCER.

*Table 1.* **Generalization to hue shift.** Classification error on the 3D Shapes datasets is reported. The models are evaluated on in-distribution (A/A), global out-of-distribution (A/B) and local out-of-distribution (A/C) test sets. $\mathbb{T}^3$CEN performs on par with LCER and achieves improved generalization performance over ResNet and CEConv.

| Network | A/A | A/B | A/C | Param |
|---|---|---|---|---|
| ResNet44 | 0.00 | 51.25 | 26.66 | 2.6M |
| CEConv-3 | 0.00 | 0.02 | 0.05 | 3.8M |
| CEConv-4 | 0.00 | 0.60 | 0.53 | 4.9M |
| LCER-H3 | 0.00 | 0.03 | 0.04 | 2.6M |
| LCER-H4 | 0.00 | **0.00** | **0.00** | 2.6M |
| $\mathbb{T}^3$CEN-H3 | 0.00 | 0.03 | 0.04 | 2.6M |
| $\mathbb{T}^3$CEN-H4 | 0.00 | **0.00** | **0.00** | 2.6M |

our lifting error is six orders of magnitude lower than LCER.

### 5.2. Generalization to Color shift

Embedding color equivariance allows for out-of-distribution (OOD) generalization. By using group actions that are compatible with the transformation space we achieve better generalization performance than existing methods.

**Hue shift.** We first evaluate our performance using the OOD hue shift experiments introduced in LCER (Yang et al., 2024). We use the 3D Shapes dataset which contains synthetically generated RGB images of 3D shapes, where the color of the shape, floor, and walls vary across examples, as does the scale and orientation of the shape (see Appendix D.1 for details). The performance of $\mathbb{T}^3$CEN, CEConv (Lengyel et al., 2023), LCER (Yang et al., 2024), and ResNet44 (He et al., 2016) are reported in Table 1. In all cases, the classification accuracy of $\mathbb{T}^3$CEN is on par or better than existing methods.

**Saturation shift.** We similarly evaluate the generalization performance of $\mathbb{T}^3$CEN, CEConv, LCER, and ResNet44 to saturation shifts. For this experiment we introduce a saturation shifted variation of the 3D Shapes dataset (see Appendix D.2 for details). The classification performance of all networks is reported in Table 2. We find that $\mathbb{T}^3$CEN has significantly better classification accuracy than all baselines.

*Table 2.* **Generalization to saturation shift.** Classification error on the saturation shifted 3D Shapes datasets is reported. The models are evaluated on in-distribution (A/A), random saturation shifts (A/B), and component-wise random saturation shifts (A/C) test sets. $\mathbb{T}^3$CEN achieve improved generalization performance over LCER and ResNet.

| Network | A/A | A/B | A/C | Param |
|---|---|---|---|---|
| ResNet44 | 0.00 | 41.40 | 42.20 | 2.6M |
| LCER-S3 | 0.00 | **0.00** | 0.04 | 2.6M |
| $\mathbb{T}^3$CEN-S3 | 0.00 | **0.00** | **0.00** | 2.6M |

*Table 3.* **Generalization to luminance shift.** Classification error on the small NORB dataset is reported. The models are evaluated on medium lighting (A/A), low lighting (A/B), and bright lighting (A/C) test sets. $\mathbb{T}^3$CEN achieves improved generalization performance over LCER and ResNet. Moreover, with the same order, $\mathbb{T}^3$CEN-L3 outperforms LCER-L3.

| Network | A/A | A/B | A/C | Parameter |
|---|---|---|---|---|
| ResNet-18 | 8.32 | 37.70 | 33.88 | 11.2M |
| LCER-L3 | 7.31 | 34.83 | 36.43 | 11.1M |
| $\mathbb{T}^3$CEN-L3 | **5.36** | 14.42 | 31.53 | 11.2M |
| $\mathbb{T}^3$CEN-L4 | 6.60 | 13.09 | 29.53 | 11.2M |
| $\mathbb{T}^3$CEN-L8 | 5.67 | 11.88 | 27.46 | 11.1M |
| $\mathbb{T}^3$CEN-L16 | 5.66 | **11.09** | **24.32** | 11.2M |

**Luminance shift.** We evaluate the generalization performance of $\mathbb{T}^3$CEN, LCER, and ResNet18 to luminance shifts on the small NORB dataset (LeCun et al., 2004) (see Appendix D.4 for details). The classification performance of all networks is reported in Table 3. We find that $\mathbb{T}^3$CEN has significantly better classification accuracy than all baselines.

**HSL shift.** We evaluate the generalization performance of $\mathbb{T}^3$CEN, LCER, and ResNet18 to HSL shifts on our HSL shifted 3D Shapes dataset (see Appendix D.3 for details). The classification performance of all networks is reported in Table 8. In contrast to baseline models, HSL-equivariant $\mathbb{T}^3$CEN-H4S4L4 achieves perfect classification accuracy.

**Color shift in the wild.** We evaluate the classification performance of $\mathbb{T}^3$CEN on Caltech-101 ( (Li et al., 2022), Oxford-IIT Pets (Parkhi et al., 2012), Stanford Cars (Krause et al., 2013), STL-10 (Coates et al., 2011), CIFAR-10, and CIFAR-100 (Krizhevsky & Hinton, 2009) (see Appendix D for details). We assess generalization performance on saturation and luminance shifted versions of these datasets in Table 6. The generalization performance of $\mathbb{T}^3$CEN consistently exceeds baseline models and augmentation techniques DeepAugment (Özmen, 2019), AugMix (Hendrycks et al., 2019), and Plankian Jitter (Zini et al., 2022).

### 5.3. Robustness to Color Imbalance.

We evaluate the robustness of $\mathbb{T}^3$CEN to color imbalance on the Camelyon17 histopathology classification dataset (Bandi et al., 2018). The Camelyon17 dataset contains images of human tissue gathered from five distinct

*Table 4.* **Generalization to HSL shift.** Classification error on the HSL shifted 3D Shapes dataset is reported. $\mathbb{T}^3$CEN achieves improved generalization performance of LCER and ResNet.

| Network | Error | Param |
|---|---|---|
| ResNet44 | 55.40 (2.19) | 2.6M |
| LCER-H4S3 | 9.76 (3.54) | 2.6M |
| $\mathbb{T}^3$CEN-H3S3L3 | 3.01 (1.61) | 2.6M |
| $\mathbb{T}^3$CEN-H4S4L4 | **0.00 (0.00)** | 2.6M |

*Table 5.* **Generalization to color shift in Camelyon17.** Classification error on the Camelyon17 dataset is reported. $\mathbb{T}^3$CEN achieves improved generalization performance over LCER and ResNet.

| Network | Error | Param |
|---|---|---|
| ResNet50 | 28.91 (7.58) | 23.5M |
| CEConv-3 | 28.76 (9.93) | 23.1M |
| LCER-H4 | 27.53 (3.39) | 23.5M |
| LCER-S3 | 16.08 (2.68) | 23.3M |
| LCER-H4S3 | 19.06 (4.92) | 23.0M |
| $\mathbb{T}^3$CEN-H4 | 27.53 (3.39) | 23.5M |
| $\mathbb{T}^3$CEN-S4 | **12.11 (2.19)** | 23.5M |
| $\mathbb{T}^3$CEN-L4 | 21.88 (2.28) | 23.5M |
| $\mathbb{T}^3$CEN-H4S4 | 20.95 (6.75) | 23.5M |
| $\mathbb{T}^3$CEN-S4L4 | 16.70 (5.44) | 23.5M |
| $\mathbb{T}^3$CEN-H4S4L4 | 19.97 (1.25) | 23.5M |

hospitals where variations arise from differences in data collection methodologies and processing procedures. Additional dataset details are provided in Appendix D.

We compare our classification performance to ResNet (He et al., 2016), CEConv, and LCER in Table 5. The saturation equivariant versions of $\mathbb{T}^3$CEN and LCER show the best performance. We attribute this to color imbalance in the dataset. Specifically, dataset statistics show that saturation varies non-uniformly across the dataset (see Appendix D.6).

In all experiments, we maintain a fixed number of network parameters across all HSL configurations. To do this, we reduce filter depth when increasing HSL order. In some cases this can lead to reduced network capacity. Consequently, networks with higher order may perform worse than networks with lower order (e.g., compare the classification accuracy of $\mathbb{T}^3$CEN-S4 and $\mathbb{T}^3$CEN-H4S4 in Table 5).

To quantify this effect, we evaluate the OOD classification performance of hue equivariant networks of different orders on the hue-shift MNIST (LeCun et al., 2002) dataset. Figure 9 in Appendix C.1 shows how classification accuracy varies with order. With increasing hue order, accuracy increases from $86.06\%$ to $98.04\%$, then decreases. Interestingly, the highest classification accuracy occurs when the hue group order has the highest entropy per element (see Section 5.4 and Figure 6 for details).

### 5.4. Double-Cover Lifting Layer

**Analyzing lifting layer coverage.** A key contribution of our method is our double-cover lifting layer. By lifting interval valued symmetries (e.g., saturation, luminance) to the circle, we can impart group structure and design group convolutional architectures that are perfectly equivariance. However, while our lifting layer covers $\mathbb{T}^1$ uniformly, how well it covers the interval is input dependent. We measure interval coverage by the entropy of the partitioning. For the interval

*Table 6.* **Generalization to color shift.** Classification error on the Caltech-101, CIFAR-10, CIFAR-100, Oxford-IIT Pets, Stanford Cars, STL-10 datasets are reported. $\mathbb{T}^3$CEN achieves improved generalization performance over baselines on saturation and luminance shifted testsets, where saturation and luminance were reduced by 0.5. Performance on the vanilla datasets are reported in Table 9 where $\mathbb{T}^3$CEN performs on par with LCER and outperforms baselines. See Appendix D for details on dataset and training.

| Network | Caltech 101 | CIFAR-10 | CIFAR-100 | Stanford Cars | Oxford Pets | STL-10 |
|---|---|---|---|---|---|---|
| **Saturation Shifted Dataset** | | | | | | |
| ResNet | 56.29 (0.59) | 11.71 (1.16) | 47.58 (5.19) | 37.34 (3.75) | 57.92 (2.49) | 30.28 (0.76) |
| ResNet-Jitter | 49.04 (0.63) | 11.91 (0.58) | 43.13 (2.13) | 32.00 (6.41) | 51.88 (1.05) | 28.86 (0.40) |
| ResNet-AugMix | **35.53 (2.50)** | 11.80 (0.61) | 46.93 (0.95) | 30.21 (0.92) | 44.84 (1.11) | 23.57 (2.36) |
| ResNet-DeepAug | 42.87 (0.47) | 20.39 (1.44) | 51.64 (0.60) | 42.32 (1.11) | 43.57 (1.65) | 24.17 (1.49) |
| ResNet-Plackian | 38.05 (2.99) | 11.87 (1.24) | 47.12 (1.89) | 33.48 (3.16) | 41.43 (4.20) | 23.69 (2.79) |
| LCER-H4 | 48.85 (1.72) | 14.28 (2.24) | 48.57 (0.35) | 31.16 (2.72) | 50.65 (4.65) | 25.05 (2.16) |
| LCER-S3 | 43.16 (1.57) | 13.28 (1.29) | 45.90 (1.08) | 40.78 (3.01) | 46.92 (0.95) | 27.53 (1.52) |
| $\mathbb{T}^3$CEN-H4 | 48.85 (1.72) | 14.28 (2.24) | 48.57 (0.35) | 31.16 (2.72) | 50.65 (4.65) | 25.05 (2.16) |
| $\mathbb{T}^3$CEN-S4 | 42.24 (1.45) | **11.59 (1.39)** | **41.35 (1.27)** | 42.91 (5.11) | 48.70 (1.46) | **21.00 (1.45)** |
| $\mathbb{T}^3$CEN-L4 | 54.00 (1.45) | 14.35 (0.57) | 49.13 (2.52) | 34.25 (8.41) | 59.55 (3.20) | 33.26 (3.44) |
| $\mathbb{T}^3$CEN-H4S4 | 40.02 (2.91) | 13.64 (0.52) | 46.23 (0.85) | 40.36 (8.34) | **38.91 (2.86)** | 22.31 (1.91) |
| $\mathbb{T}^3$CEN-S4L4 | 46.04 (5.71) | 14.13 (0.57) | 45.99 (1.70) | **29.83 (1.33)** | 54.03 (6.96) | 24.63 (5.65) |
| **Luminance Shifted Dataset** | | | | | | |
| ResNet | 73.91 (1.66) | 36.79 (3.40) | 69.39 (6.24) | 83.68 (2.96) | 80.61 (0.34) | 56.92 (1.94) |
| ResNet-Jitter | 57.03 (1.20) | 63.63 (2.33) | 59.17 (1.28) | 71.38 (5.46) | 74.05 (2.49) | 51.83 (1.19) |
| ResNet-AugMix | 57.30 (2.41) | 40.84 (1.93) | 70.60 (0.76) | 75.50 (2.52) | 74.76 (2.10) | 54.50 (0.56) |
| ResNet-DeepAug | 52.90 (1.26) | 47.86 (1.58) | 69.76 (0.61) | 75.30 (2.06) | 76.27 (2.33) | 46.58 (0.39) |
| LCER-H4 | 72.17 (2.78) | 39.81 (6.03) | 74.83 (0.87) | 76.19 (0.83) | 75.38 (1.96) | 53.45 (0.76) |
| LCER-L3 | 49.55 (2.07) | 25.62 (3.78) | **51.34 (2.16)** | 66.51 (4.57) | 64.04 (1.72) | 40.13 (1.18) |
| $\mathbb{T}^3$CEN-H4 | 72.17 (2.78) | 39.81 (6.03) | 74.83 (0.87) | 76.19 (0.83) | 75.38 (1.96) | 53.45 (0.76) |
| $\mathbb{T}^3$CEN-S4 | 70.36 (0.49) | 40.31 (6.03) | 71.39 (1.71) | 62.70 (5.17) | 79.47 (2.21) | 55.25 (2.34) |
| $\mathbb{T}^3$CEN-L4 | 50.41 (1.59) | **25.33 (1.54)** | 52.57 (1.16) | 81.42 (6.55) | 66.40 (2.82) | **38.01 (2.21)** |
| $\mathbb{T}^3$CEN-H4L4 | **46.23 (1.25)** | 39.25 (4.55) | 58.35 (1.14) | **55.55 (6.83)** | 67.97 (4.58) | 55.48 (1.07) |
| $\mathbb{T}^3$CEN-S4L4 | 50.28 (3.31) | 27.21 (0.97) | 56.71 (3.04) | 61.21 (4.18) | **62.85 (2.63)** | 41.63 (1.53) |

$(0, c)$ with the partitioning $(0, p_1, p_2, \ldots, p_{N-1}, p_N, c)$, the entropy of this partition is defined to be

$$H = \frac{1}{c} \sum_{i=1}^{N-1} v_i \log v_i, \quad v_i = p_{i+1} - p_i. \quad (18)$$

We analyze interval coverage for the case of $N = 4$ in Figure 5. Maximum coverage occurs when the input value is $kc/N - c/2N$, where $k \in \mathbb{N}$ and $k \leq N$.

**Redundant representations.** Our lifting layer produces redundant representations when the input value is $kc/N$, where $k \in \mathbb{N}$ and $k \leq N$. In these instances, our lifting layer yields repeated values

$$\left[0, \frac{c}{N}, 2\frac{c}{N}, \ldots, c, \ldots, 2\frac{c}{N}, \frac{c}{N}\right], \quad (19)$$

where the number of representations with useful information is at most $N/2 + 1$.

**Selecting the right order.** We define the best order for a particular interval to be the order with the highest average coverage $H/N$. This metric offers a principled way to select the network order using input data statistics. Figure 6 shows the average coverage across a range of orders; typically the highest average coverage is achieved at order four.

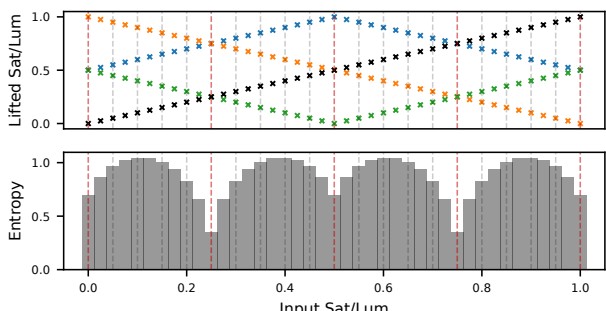

*Figure 5.* **Lifting coverage and cases of redundant representations. (Top)** Coverage of the representation $f^0 = \varphi_{hsl}(\cdot, x)$ for different input saturation and luminance values. The original input value and the first, second, and third lifted representation is in black, blue, orange, and green. **(Bottom)** Information entropy of the $f^0$. Redundant representations are highlighted with red lines.

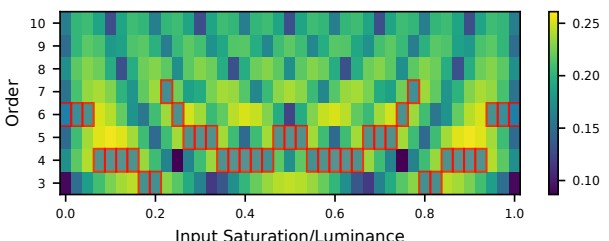

*Figure 6.* **Lifting entropy density.** We show the entropy density (information entropy divided by order) at different order and input saturation and luminance values. The order with the highest entropy density for every input is highlighted with red boxes.

## 5.5. Limitation of Color Equivariance

The preceding sections establish when $\mathbb{T}^3$CEN's structural commitments pay off. We close by examining when they do not. Color equivariance rests on two assumptions about the task: that these shifts are noise rather than signal, and that color distributions shift between training and deployment. When either assumption no longer holds, the architecture's inductive bias works against the task rather than for it. We characterize the two corresponding limitations below, and quantify each on a representative dataset.

**Color as the signal.** The first limitation occurs when absolute color predicts the class label. While equivariance embeds features across orbits of the group action, we apply an invariant pooling for classification tasks. In this setting $\mathbb{T}^3$CEN will be invariant to color information, and the network suffers reduced performance when color shift is the main feature. We show this on KUTomaData (Khan et al., 2023), a tomato ripeness dataset where color is tightly coupled to the label. Training ResNet18 and $\mathbb{T}^3$CEN for 200 epochs, $\mathbb{T}^3$CEN-H4 trails ResNet-18 by roughly 13 percentage points (see Table 7).

**No distribution shift.** The second limitation arises when training and testing data share the same color distribution. Equivariance constrains the filter bank to be tied across the group orbit, and when there is no symmetry to exploit, the constraint costs capacity without compensating generalization (see Appendix C.1). The capacity cost is structural since $\mathbb{T}^3$CEN holds total parameters fixed by reducing filter depth as lifting cardinality grows, so equivariance is paid for in expressive width. We quantify this on hue-shifted MNIST. Training ResNet44 and $\mathbb{T}^3$CEN-H4 for 1,000 iter-

*Table 7.* **Classification when color is the signal.** Classification error on KUTomaData is reported. $\mathbb{T}^3$CEN-H4 performs considerably worse than non-equivariant ResNet as the color of the tomato serves as an important signal of ripeness.

| Network | Error | Params |
|---|---|---|
| ResNet18 | 19.13 | 11.2M |
| $\mathbb{T}^3$CEN-H4 | 31.75 | 11.2M |

*Table 8.* **Classification accuracy on hue shift MNIST as a function of train-test hue shift.** We report the classification performance on hue-shifted MNIST, with increasing divergent train and test hue distributions (measured in degrees).

| Shift | $\mathbb{T}^3$CEN-H4 | ResNet44 |
|---|---|---|
| 0° | 94.18 | **98.38** |
| 5° | 95.31 | **97.94** |
| 10° | 96.10 | **97.23** |
| 15° | **97.75** | 97.06 |

ations and varying the angular shift between train and test hue. ResNet44 leads at zero shift, the gap closes as the shift grows, and $\mathbb{T}^3$CEN-H4 takes over once the distribution shift is large enough to offset the capacity cost (see Table 8).

Together, the two failure modes mark the boundary of color equivariance as an inductive bias. Specifically, $\mathbb{T}^3$CEN is suited to tasks in which color varies between training and testing and varies independently of the label.

## 6. Discussion

**Generality of the double-cover lifting.** The experimental analysis of Section 5 is specific to HSL color equivariance, but the construction it relies on is not. The double-cover lifting layer takes any interval-valued quantity and gives it the cyclic structure required for group convolution, and color is only one setting where such quantities arise. We sketch two further applications below, in RGB color space and in spatial scale, where the same construction yields cyclic group structure on a non-cyclic domain.

**Application to RGB shift equivariance.** Given the interval of valid RGB values $I = [0.0, 1.0]$, we use our lifting layer

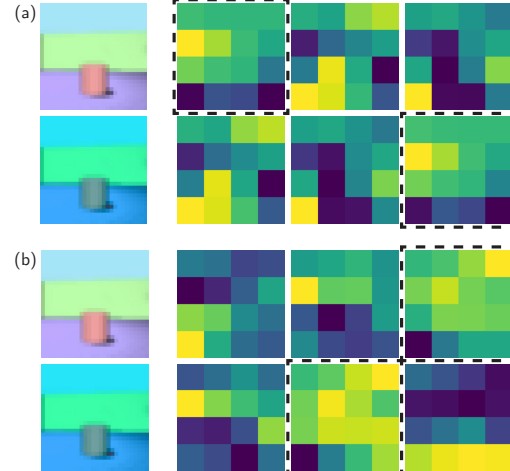

*Figure 7.* $\mathbb{T}^3$**CEN equivariant feature map under RGB shift.** To show the group actions are equivariant we qualitatively test for commutativity for both LCER and $\mathbb{T}^3$CEN. We demonstrate that the feature maps of **(a)** $\mathbb{T}^3$CEN is equivariant to shifts to the RGB channels while that of **(b)** LCER is not.

as described in Section 4 to define the $RGB$ group and group actions. Given an RGB image $x = (x_r, x_g, x_b)$, we define the group action of $g_{ijk} \in RGB_{NMR}$ on $x$ by,

$$\varphi_{rgb}(g_{ijk}, x) = \varphi_r(r_i, \varphi_g(g_j, \varphi_b(b_k, x))), \quad (20)$$

where the red, green, and blue group actions are defined as,

$$\varphi_r(r_i, x) = (\pi((x_r + r_i)(\mathrm{mod}\, 2\pi)), x_g, x_b), \quad (21)$$

$$\varphi_g(g_i, x) = (x_r, \pi((x_g + g_i)(\mathrm{mod}\, 2\pi)), x_b), \quad (22)$$

$$\varphi_b(b_i, x) = (x_r, x_g, \pi((x_b + b_i)(\mathrm{mod}\, 2\pi))). \quad (23)$$

To show the group actions are equivariant we qualitatively test for commutativity for both LCER and $\mathbb{T}^4$CEN. Figure 7 shows that a the feature maps of a red shifted input image are equal to the shifted feature maps of the original input image, thus satisfying the commutativity relationship. While this application shows it is possible to use design a color equivariant network in the RGB color space, this option may have poorer performance than a HSL color equivariant network since redundant lifted representations can be observed on all three channels.

**Application to scale equivariance.** The proposed covering map can similarly be used to design for equivariance to scale (resolution) transformations. Given the interval of valid intrinsic scale is $I = [0.0, 1.0]$ (Hosu et al., 2025), we use our lifting layer as described in Section 4 to define the scale ($\mathcal{A}$) group and group actions. Given an image with pixel location $x = (p_x, p_y)$, we define the group action of $\alpha_i \in \mathcal{A}_N$ on $x$ by,

$$\varphi_\alpha(\alpha_i, x) = \Big(\pi((p_x + g_i)(\mathrm{mod}\, 2\pi)), \quad (24)$$
$$\pi((p_y + \alpha_i)(\mathrm{mod}\, 2\pi))\Big).$$

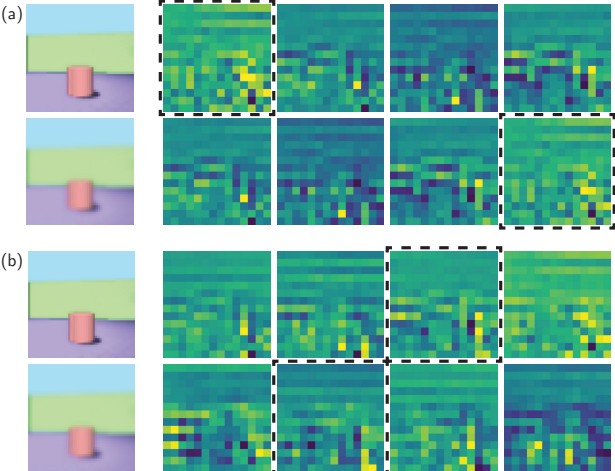

*Figure 8.* $\mathbb{T}^3$**CEN equivariant feature map under scale.** To show the group actions are equivariant we qualitatively test for commutativity for both LCER and $\mathbb{T}^3$CEN. We demonstrate that the feature maps of **(a)** $\mathbb{T}^3$CEN is equivariant to shifts in scaling while that of **(b)** LCER is not.

To show the group actions are equivariant we qualitatively test for commutativity for both LCER and $\mathbb{T}^4$CEN. Figure 8 shows that a the feature maps of a downsampled input image are equal to the shifted feature maps of the original input image, thus satisfying the commutativity relationship.

## 7. Conclusion

In this paper we introduce $\mathbb{T}^3$CEN, a hypertoroidal color equivariant network. To ensure perfect color equivariance, we use a double-cover lifting layer which can give group structure to interval valued quantities (e.g., saturation, and luminance). We show that our lifting layer preserves equivariance better than existing methods, and eliminates approximation artifacts. This translates into higher prediction accuracy in the presence of color shift, and color imbalance. We also show the broad utility of our double-cover lifting layer. Beyond the interval valued symmetries of saturation and luminance, our lifting layer can also be used to build color equivariant architectures in the RGB space, and scale equivariant architectures.

**Limitations.** The primary limitation of approach is computation expense. GCNN are typically more computationally expensive than conventional networks due to the filter orbit needed to approximate a continuous group (Cohen & Welling, 2016). The computational expense of GCNNs is roughly equal to conventional networks with a filter bank size that matches the augmented filter orbit of the GCNN.

## Impact Statement

We present $\mathbb{T}^3$CEN for perfect color equivariance. $\mathbb{T}^3$CEN is designed to achieve greater generalization to perceptual shifts, such as saturation and lighting. As $\mathbb{T}^3$CEN is design as a drop-in replacement for conventional convolutional networks, it will potentially allow for the design of more robust self-driving vision algorithms or more accurate medical imaging classification.

We also acknowledge the potential negative societal impact from fast and cheaper diagnosis, potentially leading to easier exclusion from healthcare. Additionally, as with any vision based task, it potentially allows for the advancement of surveillance and breach of privacy.

## Acknowledgements

We sincerely acknowledge the ICML 2026 reviewers and area chair for their thorough and constructive feedback. We are grateful for the thoughtful discussion and feedback from Antoine Voyer and Keqin Wang. Yulong Yang is grateful for the support and enthusiasm for this project from Dawei Yang and Feiyang Xu.

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

# A. Method

## A.1. Luminance Group and Group Action

In the HSL color space, luminance values are restricted to an interval. Because the interval does not have group structure, it is not possible to construct an GCNN on this representation directly. To get around this, (Yang et al., 2024) model luminance with the structure of the translation group $(\mathbb{R}, +)$. An element $l_i$ of resulting luminance group $L_R$, acts on HSL images by the group action

$$\varphi_l(l_i, x) = (x_h, x_s, \min(x_l + l_i, c)), \tag{25}$$

and on $L_R$ functions $f = (f_1, \ldots, f_R)$ by the group action

$$\phi_l(l_i, f) = (f_{1+i}, \ldots, f_R, \underbrace{\mathbf{0}, \ldots, \mathbf{0}}_{i}). \tag{26}$$

However, because luminance values are bounded, enforcing the translation group requires value clipping (see equation (25) and (26)). Consequently, luminance transformations can introduce spurious artifacts making learned representations only approximately equivariant.

To remedy this, we model luminance with the structure of the cyclic group $C_R$. Given the interval of valid luminance values $I = [0, c]$, we define the luminance manifold $\tilde{L}$ using the inverse of the double-cover $\pi : \mathbb{T}^1 \to I$, where $\pi(\theta) = c \sin \frac{\theta}{2}$. From there, we are able to define the luminance group $(L_R, \cdot)$. The set $L_R$ of cardinality $R$ is determined by a uniform discretization of $\tilde{L}$, and the binary operation $\cdot : L_R \times L_R \to L_R$ is defined,

$$a \cdot b \mapsto (a + b) \bmod 2\pi, \quad a, b \in L_R. \tag{27}$$

We can also define the $L_R$ group action on HSL images and functions on $L_R$. Given an HSL image $x = (x_h, x_s, x_l)$, we define the action of $l_i \in L_R$ on $x$ by,

$$\varphi_l(l_i, x) = (x_h, x_s, \pi((x_l + l_i) (\bmod 2\pi))), \tag{28}$$

and given a function $f = (f_1, \ldots, f_R)$ on the discrete saturation group $L_R$, we define the action of an element $l_i$ on $f$ by,

$$\phi_l(l_i, f) = \left(f_{(1+i)(\bmod R)}, \ldots, f_{(R+i)(\bmod R)}\right). \tag{29}$$

# B. Hue-Saturation-Luminance Group

In this section, we show that the hue, saturation, and luminance groups satisfies the axioms of a group. Recall that a *group* is defined as a set $G$ together with a binary operation $\cdot : G \times G \to G$ that satisfies the following axioms:

1. *Associativity.* For $a, b, c \in G$, $(a \cdot b) \cdot c = a \cdot (b \cdot c)$,

2. *Existence of identity.* There exists an element $e \in G$, so that for any element $a \in G$, $e \cdot a = a \cdot e = a$,

3. *Existence of inverse.* For any element $a \in G$ there exists an element $b \in G$ so that $a \cdot b = b \cdot a = e$.

## B.1. Hue Group.

Following Yang et al. (2024), we identify the elements of the discretized hue group with $H_N$ with elements of the cyclic group $C_N$. We demonstrate that the set

$$H_N = \left\{0, \ldots, \frac{2\pi k}{N}, \ldots, \frac{2\pi (N - 1)}{N}\right\}, \tag{30}$$

with integer $k$ in the range of $0 \le k < N$, along with the binary operator

$$a \cdot b \mapsto (a + b) (\bmod 2\pi), \tag{31}$$

for $a, b \in H_N$ is a group by showing that is satisfies *associativity* and has *identity* and *inverse* elements.

**Associativity.** For any hue element $a, b, c \in H_N$, defined as $a = {}^{2\pi k_a}/_N$, $b = {}^{2\pi k_b}/_N$, and $c = {}^{2\pi k_c}/_N$ where $k_a, k_b, k_c$ are integers in the range of $0 \leq k_a, k_b, k_c < N$, we can write

$$(a \cdot b) \cdot c = \left( \left( \frac{2\pi (k_a + k_b)}{N} \right) (\text{mod } 2\pi) + \frac{2\pi k_c}{N} \right) (\text{mod } 2\pi). \tag{32}$$

By recognizing that modulo operation distributes over addition such that

$$(a + b) (\text{mod } m) = (a (\text{mod } m) + b (\text{mod } m)) (\text{mod } m), \tag{33}$$

we can rewrite Equation (32) by distributing such that

$$(a \cdot b) \cdot c = \left( \left( \frac{2\pi k_a}{N} (\text{mod } 2\pi) + \frac{2\pi k_b}{N} (\text{mod } 2\pi) \right) (\text{mod } 2\pi) + \frac{2\pi k_c}{N} \right) (\text{mod } 2\pi), \tag{34}$$

$$= \left( \left( \frac{2\pi k_a}{N} (\text{mod } 2\pi) + \frac{2\pi k_b}{N} (\text{mod } 2\pi) \right) (\text{mod } 2\pi) (\text{mod } 2\pi) + \frac{2\pi k_c}{N} (\text{mod } 2\pi) \right) (\text{mod } 2\pi). \tag{35}$$

Leveraging the identity property of modulo

$$(a \bmod b) (\text{mod } b) = (a \bmod b), \tag{36}$$

we can rewrite Equation (35) such that

$$(a \cdot b) \cdot c = \left( \left( \frac{2\pi k_a}{N} (\text{mod } 2\pi) + \frac{2\pi k_b}{N} (\text{mod } 2\pi) \right) (\text{mod } 2\pi) + \left( \frac{2\pi k_c}{N} (\text{mod } 2\pi) \right) (\text{mod } 2\pi) \right) (\text{mod } 2\pi). \tag{37}$$

Using the distributive property, Equation (37) can be written as

$$(a \cdot b) \cdot c = \left( \frac{2\pi k_a}{N} (\text{mod } 2\pi) + \frac{2\pi k_b}{N} (\text{mod } 2\pi) + \frac{2\pi k_c}{N} (\text{mod } 2\pi) \right) (\text{mod } 2\pi), \tag{38}$$

$$= \left( \left( \frac{2\pi k_a}{N} (\text{mod } 2\pi) \right) (\text{mod } 2\pi) + \left( \frac{2\pi k_b}{N} (\text{mod } 2\pi) + \frac{2\pi k_c}{N} (\text{mod } 2\pi) \right) (\text{mod } 2\pi) \right) (\text{mod } 2\pi), \tag{39}$$

$$= \left( \frac{2\pi k_a}{N} (\text{mod } 2\pi) + \left( \frac{2\pi k_b}{N} (\text{mod } 2\pi) + \frac{2\pi k_c}{N} (\text{mod } 2\pi) \right) (\text{mod } 2\pi) \right) (\text{mod } 2\pi), \tag{40}$$

$$= \left( \frac{2\pi k_a}{N} + \left( \frac{2\pi (k_b + k_c)}{N} \right) (\text{mod } 2\pi) \right) (\text{mod } 2\pi), \tag{41}$$

$$= a \cdot (b \cdot c), \tag{42}$$

which shows that the hue group satisfies associativity.

**Identity element.** Consider the hue element $0 \in H_N$, then for every element $a = {}^{2\pi k_a}/_N \in H_N$ we can write

$$a \cdot 0 = \left( \frac{2\pi (k_a + 0)}{N} \right) (\text{mod } 2\pi) = a, \tag{43}$$

and

$$0 \cdot a = \left( \frac{2\pi (0 + k_a)}{N} \right) (\text{mod } 2\pi) = a, \tag{44}$$

which shows that $0$ is the identity element of $H_N$.

**Inverse element.** For any hue element $a \in H_N$, defined as $a = {}^{2\pi k_a}/_N$ where $k_a$ is an integer in the range of $0 \leq k_a, k_b, k_c < N$, an inverse element can be written as $a^{-1} = {}^{2\pi (N - k_a)}/_N$ such that

$$a \cdot a^{-1} = \left( \frac{2\pi (k_a + N - k_a)}{N} \right) (\text{mod } 2\pi) = 2\pi (\text{mod } 2\pi) = 0, \tag{45}$$

and

$$a^{-1} \cdot a = \left( \frac{2\pi (N - k_a + k_a)}{N} \right) (\text{mod } 2\pi) = 2\pi (\text{mod } 2\pi) = 0. \tag{46}$$

As $k_a$ is an integer in the range of $0 \leq k_a < N$, then $N - k_a$ is also in the range of $0 \leq N - k_a < N$, which indicates that $a^{-1} \in H_N$, which shows that $a^{-1}$ is the inverse for all hue elements $a \in H_N$.

### B.2. Saturation Group.

We identify the elements of the discretized saturation group with $S_N$ with elements of the cyclic group $C_M$. We demonstrate that the set

$$S_N = \left\{ 0, \ldots, \frac{2\pi k}{M}, \ldots, \frac{2\pi (M-1)}{M} \right\}, \tag{47}$$

with integer $k$ in the range of $0 \leq k < M$, along with the binary operator

$$a \cdot b \mapsto (a+b) \,(\mathrm{mod}\, 2\pi), \tag{48}$$

for $a, b \in S_M$. Therefore we can show that $S_M$ is a group analogously to hue in Appendix B.1.

### B.3. Luminance Group.

We identify the elements of the discretized luminance group with $L_R$ with elements of the cyclic group $C_R$. We demonstrate that the set

$$L_N = \left\{ 0, \ldots, \frac{2\pi k}{R}, \ldots, \frac{2\pi (R-1)}{R} \right\}, \tag{49}$$

with integer $k$ in the range of $0 \leq k < R$, along with the binary operator

$$a \cdot b \mapsto (a+b) \,(\mathrm{mod}\, 2\pi), \tag{50}$$

for $a, b \in L_R$. Therefore we can show that $L_R$ is a group analogously to hue in Appendix B.1.

## C. Experiments

In this section, we include additional results for Section 5.

### C.1. Degradation of Performance at Large Lifting

To facilitate fair comparison, $\mathbb{T}^3$CEN reduces the filter count when increasing total lifting cardinality (the cardinality of the hue, saturation, and luminance added together). When the order of cardinality increases, the reduced filter count may cause a decrease in classification accuracy. We demonstrate this phenomenon on a hue shifted MNIST (LeCun et al., 2002) dataset, where the training images have hue in range of $0° - 120°$ and the testing images have hue in range of $120° - 360°$. We report the classification accuracy at different cardinalities in Figure 9. For instance, the classification error with cardinality 20 is 9.19, while the lowest error of 1.96 is achieved with cardinality 4. We note that the cardinality that produced the peak accuracy matches the analytical with the highest entropy density in Figure 6. We report the corresponding final feature map channel at different cardinalities in Figure 9, where the number of channels drops from 20 to 6.

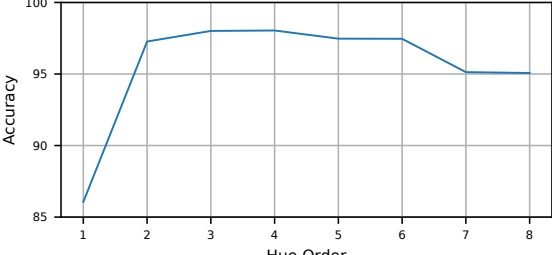 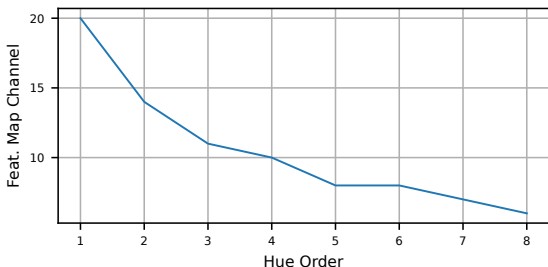

*Figure 9.* **Degradation of performance with large lifting cardinality. (Left)** We report the classification accuracy with increasing hue cardinality on the hue shift MNIST dataset (LeCun et al., 2002) (for more details see Appendix D.5). With increasing cardinality, the classification accuracy drops. **(Right)** We report the number of channels in the final feature map with increasing hue cardinality on the Z2CNN architecture (Cohen & Welling, 2016). With increasing cardinality the number of channels drops significantly.

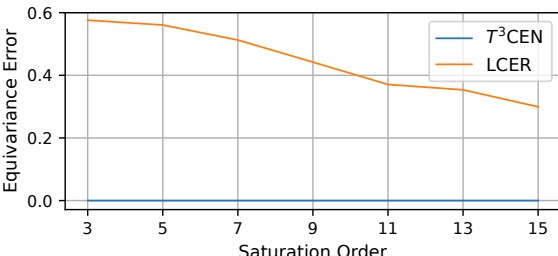

*Figure 10.* **Saturation equivariance error.** The normalized saturation equivariance error for $\mathbb{T}^3$CEN and LCER is reported. $\mathbb{T}^3$CEN has average error of $4.66 \times 10^{-6}$ while LCER has average error of $0.445$.

## C.2. Saturation Equivariance Error

Following Yang et al. (2024), we define the normalized saturation equivariance error as

$$\Delta_{\varphi_s} = \frac{|f(\varphi_s(s_i, x)) - \phi_s(s_i, f(x))|}{|f(\varphi_s(s_i, x)) + \phi_s(s_i, f(x))|}. \tag{51}$$

We calculate the normalized saturation equivariance error for LCER (Yang et al., 2024) for cardinalities of $M \in \{3, 5, 7, 9, 11, 15\}$ (as LCER use translation for lifting to the saturation group, the resulting cardinality must be odd). The saturation of the shifted input was lowered from the original input image by $^{1.0}/_{(M-1)}$ to match the lifting cardinality in LCER. We again show the equivariance error for LCER in Figure 10, where LCER exhibits significant error averaging $0.445$, while $\mathbb{T}^3$CEN has average equivariance error of $4.66 \times 10^{-6}$ (most likely due to computational error rather than architectural).

In Figure 10, we observe that the saturation equivariance error of LCER reduces with increasing cardinality. At each convolution layer after the first, LCER translates the filter orbit and zero-pads the 'missing' filter, as described in Equation (6), such that for a group action of $s_1$ the filter orbit becomes

$$\phi_s(s_1, f) = (f_2, \ldots, f_M, \mathbf{0}). \tag{52}$$

The equivariance error is, primarily, a result of the additional zero-padded filter. Intuitively, the effect of the zero-padded filter reduces with increasing cardinality (and therefore filter count). When considered in conjunction with the degradation of performance at large lifting cardinalities (see Appendix C.1 for details), decreasing equivariance error and improving classification accuracy are fundamentally opposing actions for LCER.

*Table 9.* **Generalization to color shift.** Classification error on the Caltech-101, CIFAR-10, CIFAR-100, Oxford-IIT Pets, Stanford Cars, STL-10 datasets are reported. $\mathbb{T}^3$CEN performs similarly to CEConv and LCER on the vanilla test datasets.

| | **Original Dataset** | | | | | |
|---|---|---|---|---|---|---|
| Network | Caltech 101 | CIFAR-10 | CIFAR-100 | Stanford Cars | Oxford Pets | STL-10 |
| ResNet | 32.68 (1.55) | **7.86 (1.14)** | **32.00 (0.63)** | 25.41 (0.96) | 31.52 (2.05) | 18.59 (1.65) |
| ResNet-Gray | 33.79 (3.09) | 8.45 (0.68) | 32.04 (0.66) | 24.71 (0.93) | 30.38 (0.35) | 18.71 (1.47) |
| ResNet-Jitter | 32.90 (0.82) | 8.33 (0.44) | 32.27 (0.18) | 22.38 (1.65) | 30.06 (0.52) | **17.89 (1.48)** |
| CEConv-3 | 34.74 (0.83) | 8.86 (0.33) | 34.95 (0.44) | 23.97 (1.56) | 31.08 (2.54) | 24.29 (1.31) |
| CEConv-4 | 33.52 (0.48) | 9.28 (0.24) | 35.46 (0.35) | 24.08 (0.66) | 33.70 (1.50) | 21.90 (1.64) |
| LCER-H4 | **32.23 (1.07)** | 8.83 (0.64) | 34.70 (0.89) | 20.38 (1.06) | **27.39 (0.68)** | 20.73 (1.11) |
| LCER-S3 | 41.64 (1.40) | 9.24 (0.27) | 39.33 (0.45) | 31.90 (10.03) | 36.87 (5.57) | 20.71 (1.10) |
| LCER-H4S3 | 38.14 (1.07) | 10.68 (0.78) | 33.27 (0.31) | 24.79 (3.87) | 29.84 (1.34) | 20.53 (0.73) |
| $\mathbb{T}^3$CEN-H4 | **32.23 (1.07)** | 8.83 (0.64) | 34.70 (0.89) | 20.38 (1.06) | **27.39 (0.68)** | 20.73 (1.11) |
| $\mathbb{T}^3$CEN-S4 | 36.67 (0.65) | 9.38 (0.84) | 35.08 (0.72) | 23.46 (2.95) | 35.53 (2.91) | 21.50 (1.18) |
| $\mathbb{T}^3$CEN-L4 | 37.38 (1.18) | 9.46 (1.24) | 34.74 (0.53) | **20.33 (2.20)** | 41.30 (0.33) | 22.77 (0.47) |
| $\mathbb{T}^3$CEN-H4S4 | 36.58 (0.36) | 11.04 (0.90) | 40.87 (0.52) | 24.98 (0.70) | 35.44 (1.03) | 21.63 (0.41) |
| $\mathbb{T}^3$CEN-S4L4 | 37.42 (1.09) | 11.18 (0.57) | 39.32 (0.27) | 25.84 (0.28) | 39.94 (4.09) | 22.53 (1.06) |

*Table 10.* **Redundant lifting metric.** We report the percentage of pixels in the datasets in Table 6 that exhibit full or partial redundant values.

|  | Caltech 101 | CIFAR-10 | CIFAR-100 | Stanford Cars | Oxford Pets | STL-10 |
|---|---|---|---|---|---|---|
| Full | 0.80% | 0.57% | 0.69% | 0.88% | 0.65% | 0.89% |
| Partial | 0.69% | 0.64% | 0.64% | 0.67% | 0.48% | 0.66% |

### C.3. Expanded Results Table

We report the classification accuracies on the vanilla datasets in Table 9.

### C.4. Redundant Lifting

Redundant cases occur when the input value is $^{kc}/_N$, where $k \in \mathbb{N}$ and $k \leq N$. In this case, the lifting will yield at most $^N/_2 + 1$ representations with useful information. We define fully redundant inputs as those that have value $0.5$ and partially redundant inputs as those that have value $0.25$ or $0.75$. We quantify the effects of these redundant lifting on the datasets in Table 6 in Table 10.

## D. Dataset and Implementation Details

In this section, we include details on the dataset and network implementation for results presented in Section 5.

All experiments were performed over a set of random seeds to evaluate the robustness of $\mathbb{T}^3$CEN to initialization. All errors reported for 3D Shapes, Small NORB, CIFAR-10, CIFAR-100, Caltech-101, Oxford Pets, Stanford Cars, and STL-10 are based on three random seeds (1999-2001). The error reported for Camelyon17 were based on five random seeds (1997-2002).

All models were trained on a shared research computing cluster, where each GPU enabled compute node was allocated an Nvidia L40 GPU, 24 core partitions of an Intel Xeon Gold 5320 CPU, and 24GBs of DDR4 3200MHz RDIMMs. We report training times and memory requirements in Appendix D.8

We report training hyperparameters for Caltech-101, CIFAR-10, CIFAR-100, Oxford-IIT Pets, Stanford Cars, STL-10 in Table 11. We report training procedure for hue shift 3DShapes, saturation shift 3DShapes, small NORB, and Camelyon17 in their individual sections.

### D.1. Hue Shift 3D Shapes

We evaluate the hue-equivariance performance of $\mathbb{T}^3$CEN on the hue shift 3D Shapes (Krause et al., 2013). The train set and in-distribution test set ($A$) consists of images where the floor, background, and object all have warm hue (indexed by 0-4 in the original 3D Shapes dataset); the first out-of-distribution test set ($B$) consists of images where the floor, background, and object all have cold hue (indexed by 5-9 in the original 3D Shapes dataset); and the second out-of-distribution test set ($C$) consists of images where the floor and background have warm hue and the object has cold hue. Example images from all three subsets are shown in Figure 11a.

*Table 11.* **Training hyperparameters.** We report the training hyperparameters on the Caltech-101, CIFAR-10, CIFAR-100, Oxford-IIT Pets, Stanford Cars, STL-10 datasets. All datasets are trained under cross entropy loss. All datasets are training by replacing conventional convolution blocks with $\mathbb{T}^3$CEN convolution. The filter count for $\mathbb{T}^3$CEN is reduced to match that of the respective vanilla ResNet. Group pooling is performed at the last layer to obtain an invariance representation for classification. LCER and CEConv are training using network parameters provided in Lengyel et al. (2023) and Yang et al. (2024) respectively.

|  | Caltech 101 | CIFAR-10 | CIFAR-100 | Stanford Cars | Oxford Pets | STL-10 |
|---|---|---|---|---|---|---|
| Arch. | ResNet18 | ResNet44 | ResNet44 | ResNet18 | ResNet18 | ResNet18 |
| Batch Size | 16 | 128 | 128 | 16 | 16 | 16 |
| Epoch | 300 | 300 | 300 | 300 | 300 | 300 |
| Optimizer | Adam | SGD | SGD | Adam | Adam | Adam |
| Learning Rate | $10^{-2}$ | $10^{-1}$ | $10^{-1}$ | $10^{-2}$ | $10^{-2}$ | $10^{-2}$ |
| Scheduler | N/A | cos anneal. | cos anneal. | N/A | N/A | N/A |

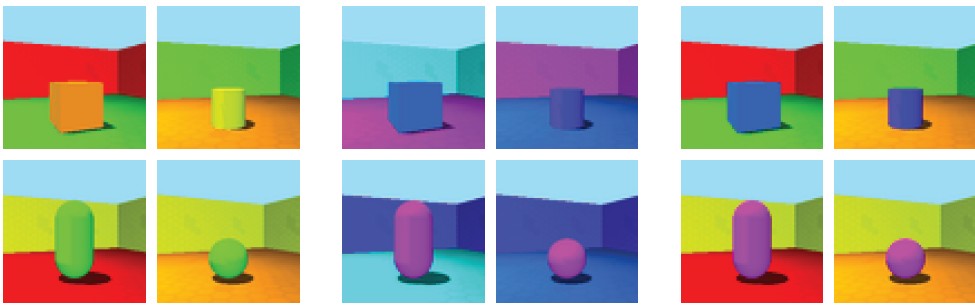

*(a)* **Hue shift 3D Shapes dataset. (Left)** The train set and in-distribution test set $A$, where the floor, background, and object all have warm hues (colors 0-4); **(Middle)** The first out-of-distribution test set $B$, where the floor, background, and object all have cold hues (colors 5-9); **(Right)** The second out-of-distribution test set $C$, where the floor and background have warm hues and the object has cold hues.

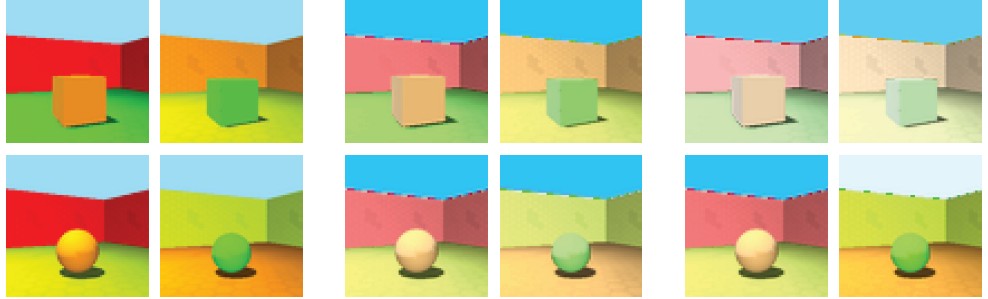

*(b)* **Saturation shift 3D Shapes dataset. (Left)** The train set and in-distribution test set $A$ with original saturation values; **(Middle)** The first out-of-distribution test set $B$ with regular saturation shifts of multiples of 0.125; **(Right)** The second out-of-distribution test set $C$ with saturation shifts of random values.

*Figure 11.* **Hue and saturation shifted 3D shapes.** Examples images from the **(a)** hue and **(b)** saturation shifted 3D shapes dataset.

We compare the classification accuracy of hue-equivariant $\mathbb{T}^3$CEN with ResNet44, CEConv (Lengyel et al., 2023), and hue-equivariant LCER (Yang et al., 2024). $\mathbb{T}^3$CEN has the same architectural backbone as ResNet44, but with a reduced filter count to ensure similar number of network parameters. We perform group pooling at the last layer of $\mathbb{T}^3$CEN to obtain a hue-invariant representation for classification. We optimized $\mathbb{T}^3$CEN under cross-entropy loss using Adam (Kingma & Ba, 2015) with learning rate of $10^{-4}$ for $10,000$ iterations with a batch size of 128.

### D.2. Saturation Shift 3D Shapes

We evaluate the saturation-equivariance performance of $\mathbb{T}^3$CEN on the saturation shift 3D Shapes (Krause et al., 2013). The train set and in-distribution test set ($A$) consists of images with the original saturation; the first out-of-distribution test set ($B$) consists of images with regular saturation shifts of multiples of $0.125$; and the second out-of-distribution test set ($C$) consists of images with saturation shifts of random values. Example images from all three subsets are shown in Figure 11b.

We compare the classification accuracy of saturation-equivariant $\mathbb{T}^3$CEN with ResNet44 and saturation-equivariant LCER (Yang et al., 2024). $\mathbb{T}^3$CEN has the same architectural backbone as ResNet44, but with a reduced filter count to ensure similar number of network parameters. We perform group pooling at the last layer of $\mathbb{T}^3$CEN to obtain a saturation-invariant representation for classification. We optimized $\mathbb{T}^3$CEN under cross-entropy loss using Adam (Kingma & Ba, 2015) with learning rate of $10^{-4}$ for $10,000$ iterations with a batch size of 128.

### D.3. Hue-Saturation-Luminance Shift 3D Shapes

We evaluate the HSL-equivariance performance of $\mathbb{T}^3$CEN on the HSL shift 3D Shapes (Krause et al., 2013). The train set and in-distribution test set ($A$) consists of images with the original hue, saturation, and luminance; the test set consists of images with random shifts in HSL.

We compare the classification accuracy of HSL-equivariant $\mathbb{T}^3$CEN with ResNet44. $\mathbb{T}^3$CEN has the same architectural backbone as ResNet44, but with a reduced filter count to ensure similar number of network parameters. We perform group

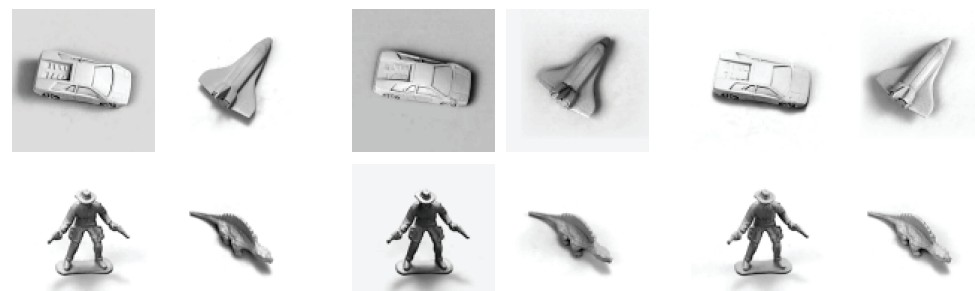

*Figure 12.* **Small NORB dataset. (Left)** The train set and in-distribution test set $A$ with medium lighting conditions; **(Middle)** The first out-of-distribution test set $B$ with low lighting conditions; **(Right)** The second out-of-distribution test set $C$ with high lighting conditions.

pooling at the last layer of $\mathbb{T}^3$CEN to obtain a HSL-invariant representation for classification. We optimized $\mathbb{T}^3$CEN under cross-entropy loss using Adam (Kingma & Ba, 2015) with learning rate of $10^{-4}$ for 10k iterations with a batch size of 128.

### D.4. Small NORB

We evaluate the performance of $\mathbb{T}^3$CEN in the presence of lighting shifts on the small NORB (LeCun et al., 2004) dataset, which consists of 48.6k $96 \times 96$ resolution grayscale images rendered under 6 lighting conditions, 9 elevations, and 18 azimuths. For ease of processing (as the original dataset was stored using MATLAB files), we use the pre-processed dataset given in `https://huggingface.co/datasets/Ramos-Ramos/smallnorb`. The train set and in-distribution test set ($A$) consists of images with medium lighting conditions (2-3 as defined in LeCun et al. (2004)); the first out-of-distribution test set ($B$) consists of images with low lighting conditions (0-1 as defined in LeCun et al. (2004)); and the second out-of-distribution test set ($C$) consists of images with high lighting conditions (4-5 as defined in LeCun et al. (2004)). Example images from all three subsets are shown in Figure 12.

We compare the classification accuracy of luminance-equivariant $\mathbb{T}^3$CEN with ResNet18 and luminance-equivariant LCER (Yang et al., 2024). $\mathbb{T}^3$CEN has the same architectural backbone as ResNet18, but with a reduced filter count to ensure similar number of network parameters. We perform group pooling at the last layer of $\mathbb{T}^3$CEN to obtain a luminance-invariant representation for classification. We optimized $\mathbb{T}^3$CEN under cross-entropy loss using Adam (Kingma & Ba, 2015) with learning rate of $10^{-2}$ for 300 epochs with a batch size of 16.

### D.5. Hue Shift MNIST

We evaluate the performance of $\mathbb{T}^3$CEN with at different cardinalities using the hue shift MNIST (LeCun et al., 2002) dataset. The train set includes hue in the range of $0° - 120°$ and the test set includes hue in the range of $120° - 360°$. Example images are shown in Figure

We compare the classification accuracy of $\mathbb{T}^3$CEN at with hue lifted to different cardinalities. $\mathbb{T}^3$CEN has the same architectural backbone as Z2CNN (Cohen & Welling, 2016), but with a reduced filter count to ensure similar number of network parameters. We perform group pooling at the last layer of $\mathbb{T}^3$CEN to obtain a hue-invariant representation for classification. We optimized $\mathbb{T}^3$CEN under cross-entropy loss using SGD with learning rate of $10^{-3}$ for 5 epochs with a

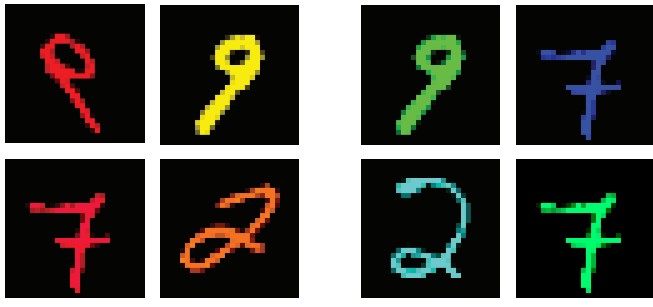

*Figure 13.* **Hue shift MNIST dataset. (Left)** The train set hue in range of $0° - 120°$ and **(Right)** the test set hue in range of $120° - 360°$.

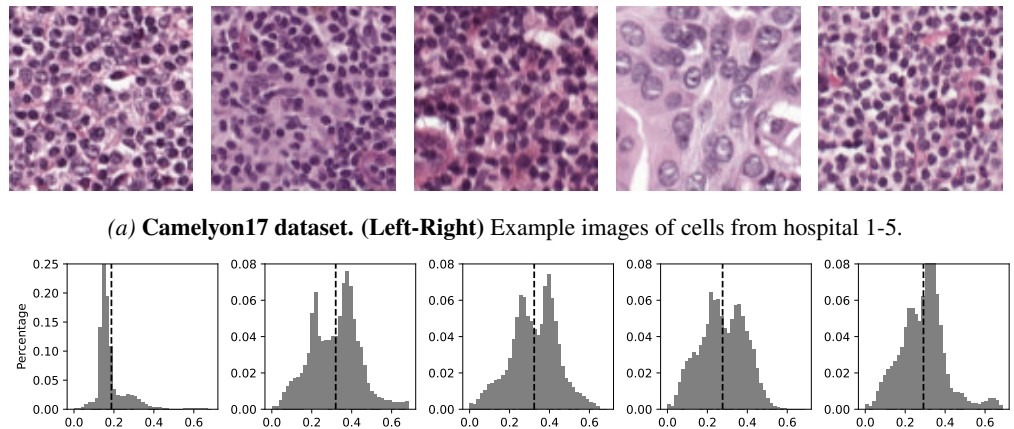

*(a)* **Camelyon17 dataset. (Left-Right)** Example images of cells from hospital 1-5.

*(b)* **Camelyon17 dataset saturation distribution. (Left-Right)** Saturation distribution of images from hospital 1-5.

*Figure 14.* **Camelyon17 dataset.** We show **(a)** example slides from the Camelyon17 dataset as well as **(b)** saturation distribution.

batch size of 128.

### D.6. Camelyon17

We evaluate the performance of $\mathbb{T}^3$CEN on medical imaging on the Camelyon17 (Bandi et al., 2018) dataset. The dataset consists of Whole-Slide Images (WSI) of Hematoxylin and Eosin (H&E) stained lymph node sections. For each patient, five slides are provided. A total of 100 patient slide imaging is released, with 50 for training and 50 for testing. The training dataset consists of all imaged from hospital 0, 3, and 4, the validation dataset consists of all images from hospital 2, and the testing dataset consists of all images from hospital 1. Example cell images from all five hospitals are shown in Figure 14a. We show the distribution of image saturation values in Figure 14b.

We compare the classification accuracy of $\mathbb{T}^3$CEN with ResNet50, CEConv (Lengyel et al., 2023), and LCER (Yang et al., 2024). $\mathbb{T}^3$CEN has the same architectural backbone as ResNet50, but with a reduced filter count to ensure similar number of network parameters. We perform group pooling at the last layer of $\mathbb{T}^3$CEN to obtain a invariant representation for classification. We optimized $\mathbb{T}^3$CEN under cross-entropy loss using Adam (Kingma & Ba, 2015) with learning rate of $10^{-2}$. We trained $\mathbb{T}^3$CEN for $10,000$ iterations with a batch size of 32.

### D.7. KUTomaData

One failure mode for enforcing color equivariance arises when absolute color predicts the class label. Equivariance pools features across orbits of the group action, so when the discriminative signal lives along these orbits, the architecture suppresses what it should preserve. We test this on KUTomaData (Khan et al., 2023), a binary tomato ripeness benchmark in which color is tightly coupled to the ripe/unripe label. Example images are shown in Figure 15.

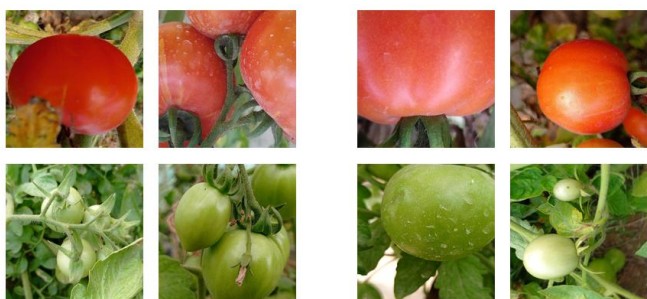

*Figure 15.* **KUTomaData dataset.** Examples images from the KUTomaData dataset, showing both ripe and unripe tomatoes in the train and test partition.

*Table 12.* **Training computation expense.** We report the training iterations per second and memory consumption, where $\mathbb{T}^3$CEN. Training memory consumption is obtained from `nvidia-smi`. Training time is reported using `tqdm`. For both metrics we discard the first 100 iterations to account for model loading. We report the value averaged over the next 100 iterations for the ResNet18 backbone. LCER does not support lifting to both the saturation and luminance channel.

| | H1 | H4 | S4 | L4 | H4S4 | H4L4 | H4S4L4 |
|---|---|---|---|---|---|---|---|
| | | | Training Memory (MiB) | | | | |
| $\mathbb{T}^3$CEN | 1030 | 2739 | 2739 | 2739 | 9480 | 9480 | 32842 |
| LCER | 1014 | 2702 | 2702 | 2702 | 9906 | 9906 | not possible |
| | | | Training Speed (it/s) | | | | |
| $\mathbb{T}^3$CEN | 19.38 | 11.33 | 11.34 | 11.27 | 5.23 | 5.24 | 1.59 |
| LCER | 19.01 | 11.30 | 11.31 | 11.31 | 5.22 | 5.21 | not possible |

## D.8. Computational Requirement

We report the training iterations per second and memory usage in Table 12 below. Training memory consumption is obtained from `nvidia-smi`. Training time is reported using `tqdm` (da Costa-Luis, 2019). For both metrics we discard the first 100 iterations to account for model loading. We report the value averaged over the next 100 iterations for the ResNet18 backbone.

