# OpenReview forum: "A Hypertoroidal Covering for Perfect Color Equivariance"
_ICML.cc/2026/Conference — ICML 2026 regular_

### Official Review · Reviewer_qJ2k · 2026-03-02

**Soundness:** 3
**Presentation:** 4
**Significance:** 3
**Originality:** 3
**Overall Recommendation:** 5
**Confidence:** 4

**Summary:**

This paper proposes hypertoroidal color equivariant network, a color equivariant architecture that is perfectly equivariant to shifts in hue, saturation, and luminance. The authors attempts to analyze a central topic: prior color equivariant methods model saturation and luminance as 1D translations on the interval, which introduces clipping and approximation artifacts; the authors consider the concept of using a topological double-cover to lift the interval to the circle (T¹), imparting cyclic group structure so that group convolutions can be defined and equivariance is exact. The double-cover is given by π(θ) = (c/2) sin θ for saturation (and similarly for luminance), with with discrete cyclic groups $S_M$ (saturation) and $L_R$ (luminance). An HSL lifting layer and HSL group convolution yield perfect equivariance; the paper also shows application of the same lifting idea to RGB shift and to scale equivariance. Experiments on synthetic (3D Shapes, hue/saturation/luminance/HSL shift) and datasets like Camelyon17 and others show large reductions in saturation equivariance error lower lifting error, and better OOD generalization to color shift and color imbalance. The paper includes a principled order-selection criterion (entropy-based coverage), discusses degenerate lifting, and acknowledges computational cost.

**Compliance With Llm Reviewing Policy:**

Affirmed.

**Final Justification:**

Accept

Justification:
- The authors responded during the rebuttal phase and presented the empirical evaluation which is sufficient to support the main claims.
- The authors did provide additional evidence to address these concerns and acknowledged them to be added in the paper.

**Key Questions For Authors:**

- How often do degenerate lifting cases occur on your real datasets (e.g., Camelyon17, CIFAR)? Do you have mitigation strategies or preprocessing to reduce their impact?
- Can you report training/inference time, FLOPs, or memory for $T^3$CEN vs. LCER vs. ResNet at comparable depth? How does cost scale with N, M, R?

**Limitations:**

yes

**Strengths And Weaknesses:**

Strengths

1. **Novel double-cover lifting for perfect color equivariance.** Replacing the interval-as-translation approximation (LCER) with a double-cover π : T¹ → Ĩ (e.g., π(θ) = (c/2) sin θ) gives cyclic group structure to saturation and luminance, enabling exact equivariance without clipping. The construction is clearly presented (Sec 4, Appendix A.1, B), and the equivariance of the lifting layer and HSL group convolution is established. This addresses a real limitation of prior work (approximation artifacts, lifting error).

2. **Strong empirical validation.** Saturation equivariance error drops from ~0.445 (LCER) to ~4.66×10⁻⁶ ($T^3$CEN ) across group orders (Figure 3); lifting error is orders of magnitude lower (Figure 4). $T^3$CEN matches or outperforms LCER and ResNet on hue shift (Table 1), outperforms on saturation shift (Table 2), luminance shift (Table 3), HSL shift (Table 4), Camelyon17 (Table 5), and saturation/luminance-shifted natural image datasets (Table 6). Results are reported with standard deviations over multiple seeds (Appendix D), improving reproducibility.

3. **Principled analysis of the lifting layer.** Section 5.4 analyzes interval coverage via entropy (Eq. 18), identifies degenerate lifting when input is kc/N (Eq. 19), and proposes selecting group order by highest average coverage H/N (Figure 6). This gives practical guidance and explains why very high orders can hurt (Appendix C.1: capacity vs. cardinality trade-off).

4. **Broader applicability of the covering.** The same double-cover idea is applied to RGB shift equivariance (Section 5, Figure 7) and scale equivariance (Figure 8), demonstrating generality beyond HSL. The paper briefly notes that RGB may suffer from degenerate lifts on all three channels compared to HSL.

5. **Relevance to real-world problems.** Evaluation on Camelyon17 (histopathology, color imbalance across hospitals) and on saturation/luminance-shifted natural image benchmarks addresses robustness in medical imaging and fine-grained recognition. The impact statement discusses both positive (robust vision, medical imaging) and negative societal impact (healthcare exclusion, surveillance/privacy).

**Weaknesses**

1. **Limited comparison with strong augmentation baselines.** The paper compares $T^3$CEN  to ResNet, ResNet-aug, CEConv, and LCER. It does not provide a systematic, compute- or parameter-matched comparison with a strong conventional network trained with heavy color augmentation (e.g., broad saturation/luminance/hue augmentation). Such a comparison would clarify when structural equivariance wins vs. data augmentation.

2. **Degenerate lifting in practice.** The paper identifies degenerate cases (input kc/N yields repeated lifted values, at most N/2+1 useful representations) and notes that interval coverage is input-dependent (Figure 5). It does not quantify how often degenerate lifting occurs in real datasets or how much it affects accuracy; nor does it propose mitigation (e.g., input preprocessing or order selection to avoid degeneracy).

3. **Computational cost not quantified.** The conclusion mentions that GCNNs are more expensive due to filter orbits and that cost is “roughly equal to conventional networks with a filter bank size that matches the augmented filter orbit.” The paper does not report training/inference time, FLOPs, or memory for $T^3$CEN vs. LCER vs. ResNet, or how cost scales with HSL order (N, M, R).

---

> ### Author Rebuttal · Authors · 2026-03-31
>
> Thank you for your time and the detailed and insightful suggestions. We are motivated that you found T3CEN to be novel, performant, and relavant to real world applications. We also appreciate your concerns and questions, specifically those focusing on stronger baseline comparison, systematic analysis of degenerate representations, and computational cost. We hope to address these in our rebuttal below, and will incorporate all feedback in our revision.
>
> >W1. Limited comparison w/...augmentation baselines.
>
> Thank you for encouraging us to expand our evaluation. We have included additional comparisons with ResNet trained using AugMix [1], DeepAugment [2], and Planckian Jitter [3], reported below. Notably Planckian Jitter focuses on systematic color augmentation that produces photo-realistic representations. Across these stronger augmentation baselines, T3CEN outperforms augmentation-based approaches on 11 of the 12 shifted benchmarks. These additional results show that explicitly modeling color geometry, as in T3CEN, can provide consistent empirical gains beyond augmentation alone. In all experiments (in the rebuttal and the paper), we keep the backbone architecture fixed and maintain the same parameter budget across models. We achieve this by reducing the filter depth of T3CEN with increasing HSL order (see Appendix C.1).
>
> |Sat-Shift|Caltech|CIFAR-10|CIFAR-100|Cars|Pets|STL|
> |-|:-:|:-:|:-:|:-:|:-:|:-:|
> |ResNet-augmix|35.53|11.80|46.93|30.21|44.84|23.57|
> |ResNet-DeepAugment|42.87|20.39|51.64|42.32|43.57|24.17|
> |ResNet-PlanckianJitter|38.81|12.28|51.33|31.25|41.43|21.83|
> |T3CEN|40.02|11.59|44.27|29.83|38.91|21.00|
>
> |Lum-Shift|Caltech|CIFAR-10|CIFAR-100|Cars|Pets|STL|
> |-|:-:|:-:|:-:|:-:|:-:|:-:|
> |ResNet-augmix|57.30|40.84|70.60|75.50|74.76|54.50|
> |ResNet-DeepAugment|52.90|47.86|69.76|75.30|76.27|46.58|
> |T3CEN|46.23|25.33|52.57|55.55|62.85|38.01|
>
> Due to the rebuttal time limit, we only compare with Planckian Jitter on the saturation shifted dataset. We will round out the evaluation with  luminance-shift in our revision and expand our discussion and evaluation to these augmentation baselines.
>
> [1] Dan Hendrycks, et al. ICLR, 2020. [2] Dan Hendrycks, et al. ICCV, 2021. [3] Simone Zini, et al. ICLR, 2023.
>
> >W2/Q1. Degenerate lifting in practice.
>
> We do not believe that these degenerate lifted representations have a significant practical impact. For the case shown in Figure 5, fully degenerate representations occur only when the input pixel has saturation or luminance value 0.5, and partially degenerate representations occur at 0.25 and 0.75. We show below the percentage of pixels that yield  degenerate representations for the datasets studies in Table 6.
>
> |Dataset|Fully Degen. (S=0.5)|Partially Degen. (S=0.25/0.75)|
> |:-:|:-:|:-:|
> |CIFAR-10|0.80%|0.69%|
> |CIFAR-100|0.57%|0.64%|
> |Caltech-101|0.69%|0.64%|
> |Oxford Pets|0.88%|0.67%|
> |Stanford Cars|0.65%|0.48%|
> |STL-10|0.89%|0.66%|
>
> Moreover, Figure 6 suggests a practical way to mitigate this effect: selecting the lifting order with the highest information entropy based on the input HSL distribution. We will clarify this point in the revision and include quantitative measurements of degenerate representations for all datasets studied.
>
> To quantify the effect of degenerate representations on performance, we train T3CEN-S4 on a preprocessed CIFAR-100 dataset. We apply a small non-zero saturation jitter to input pixels with values 0.25, 0.5, and 0.75 to remove degenerate representations. We find that T3CEN performs similarly on the original and preprocessed datasets. We will include this comparison, along with variance, in the revision.
>
> ||Original|Preprocessed|
> |-|:-:|:-:|
> |T3CEN-S4|44.27|44.64|
>
> Thank you for raising this important issue and pushing us to better quantify the limitations of T3CEN. We will include these results and expand our discussion in our revision.
>
> >W3/Q2. Computational cost not quantified.
>
> Thank you for pointing out this omission. We have included training times and memory consumption for T3CEN, LCER, and ResNet below. As with LCER, T3CEN introduces added compute and memory overhead, which is a standard tradeoff in group-equivariant convolutions [1]. We note that H1 reduces to ResNet.
>
> ||H1|H4|S4|L4|H4S4|H4L4|H4S4L4|
> |:-|:-:|:-:|:-:|:-:|:-:|:-:|:-:|
> |MiB-T3CEN|1030|2739|2739|2739|9480|9480|32842|
> |MiB-LCER|1014|2702|2702|2702|9906|9906|not possible|
> |iter/s-T3CEN|19.38|11.33|11.34|11.27|5.23|5.23|1.58|
> |iter/s-LCER|19.01|11.30|11.31|11.31|5.22|5.21|not possible|
>
> The additional cost becomes more pronounced for higher-order joint HSL lifting. At comparable orders, T3CEN and LCER have very similar memory usage and training throughput, while T3CEN supports H4S4L4, which is not feasible in LCER. The empirical evaluation in Table 6 indicates that T3CEN uses the additional compute more effectively compared to LCER. We will add this comparison to the revision to make the cost-performance tradeoff more explicit.
>
> [1] Taco Cohen and Max Welling. ICML, 2016.

---

> > ### Author Rebuttal · Reviewer_qJ2k · 2026-04-03
> >
> > Thank you for the detailed rebuttal and the additional experiments, it did address some of my concerns. My review also highlighted quantitative evaluation for scale and RGB extensions, a clearer recipe for choosing (N, M, R) beyond coverage entropy, and guidance on when full H+S+L helps versus partial equivariance on Camelyon17.  Therefore, I will maintain my original score.

---

> > > ### Author Response · Authors · 2026-04-05
> > >
> > > Thank you for your reply and the opportunity to extend the discussion. We will address you remaining concerns below.
> > >
> > > > When full H+S+L helps on Camelyon17.
> > >
> > > On Camelyon17, the benefit of full H+S+L versus partial equivariance depends on whether we enforce parameter count parity with baselines. Full HSL equivariance achieves the best performance when parameter count is unconstrained; single channel saturation-equivariance produces the strongest performance when adhering to a parameter budget.
> > >
> > > **Unconstrained Parameter Count.** When we do not reduce network width, and allow model capacity to grow with lifting order, higher-order equivariance becomes beneficial. We denote unconstrained parameter T3CEN with suffix “-UC”. On ResNet50 backbone, H4S4-UC improves over S4, and S4L4-UC improves further.
> > >
> > > |ResNet50|T3CEN-S4|T3CEN-H4S4|T3CEN-H4S4-UC|T3CEN-S4L4-UC|T3CEN-H4S4L4-UC|
> > > |-|:-:|:-:|:-:|:-:|:-:|
> > > |Error|12.11|20.95|11.63|9.77|n/a|
> > > |Param.|23.5M|23.5M|93.8M|93.8M|375.2M|
> > >
> > > These improvements are consistent with observed color shift in Camelyon (see https://imgur.com/a/DDx2I4c), which is dominated by sat and lum. In particular, the average HSL shift between the train and validation slides is
> > > |$\Delta$ Hue|$\Delta$ Saturation|$\Delta$ Luminance|
> > > |:-:|:-:|:-:|
> > > |0.010|0.060|0.036|
> > >
> > > Although we were unable to run H4S4L4-UC on ResNet50 within the rebuttal timeline due to compute limits, results on the smaller ResNet44 backbone suggest the same trend continues, with higher-order equivariance continuing to help in the unconstrained regime. Notably full HSL equivariance achieves the strongest performance.
> > >
> > > |ResNet44|T3CEN-S4|T3CEN-H4S4-UC|T3CEN-S4L4-UC|T3CEN-H4S4L4-UC|
> > > |-|:-:|:-:|:-:|:-:|
> > > |Error|37.07|33.87|27.74|26.01|
> > >
> > > **Constrained Parameter Count** (default setting in the paper). When we keep the parameter budget fixed by reducing network width as lifting order increases, partial equivariance is preferable. In Table 5, T3CEN-S4 achieves the best error (12.11), while adding additional hue or lum equivariance worsens performance (20.95 for H4S4, 16.70 for S4L4, and 19.97 for H4S4L4). Under a fixed parameter budget, the extra channels are therefore not always worth the accompanying reduction in backbone capacity. We will clarify this tradeoff in the revision and connect it more explicitly to our discussion on order selection below.
> > >
> > > > Choosing (N, M, R).
> > >
> > > Choosing (N,M,R) in the parameter-constrained setting involves two decisions: selecting the active HSL subgroup, and then selecting the order within each active channel. Our recommended procedure is:
> > > 1. Estimate per-channel shift. Measure the train-validation distribution shift for hue, sat, and lum using dataset statistics. The channel with the strongest shift should be the main candidate for equivariance.
> > > 2. Choose the active subgroup. Start with a single-channel T3CEN on the dominant shifted channel, then add additional channels only when they also exhibit meaningful shift and improve validation accuracy.
> > > 3. Choose per-channel order. For each channel, restrict attention to orders with high entropy density (see Figure 6). Empirically, order 4 is a robust default for sat/lum in our experiments.
> > > 4. Account for the parameter budget. Include only channels that have significant distribution-shifts to justify enforcing equivariance. Start with a single channel and work your way up to find the tradeoff between higher-order equivariance and expressive capacity.
> > > 5. Check computational feasibility. The final (N,M,R) should satisfy the available memory and training-time budget.
> > >
> > > **Camelyon17 Example.** On Camelyon17, the dominant train-validation shift is in saturation, with a smaller luminance shift and minimal hue shift. This suggests starting from a saturation-equivariant model. Entropy-density analysis identifies order 4 as a strong default for Camelyon's saturation values [0, ~0.65] (see https://imgur.com/a/DDx2I4c). Among parameter-matched models T3CEN-S4 achieves the best validation/test performance. Although lum also shifts, adding lum-equivariance does not improve performance enough to offset the reduced backbone capacity under a fixed parameter budget.
> > >
> > > > Scale and RGB extensions.
> > >
> > > Thank you for raising this question. We included a brief discussion on scale equivariance to demonstrate that our double cover can be used to resolve approximation artifacts beyond color. Hence, as scale equivariance is not a property of the proposed T3CEN architecture, we did not include a comprehensive comparison with dedicated scale-equivariant architectures such as [2,3]. We will move the discussion to the appendix and indicate that scale is an proof-of-concept extension of our lifting. Thank you for highlighting areas in our paper that may cause confusion.
> > >
> > > [2] Ivan Sosnovik, et al. ICLR, 2020.\
> > > [3] Md Ashiqur Rahman and Raymond A. Yeh. NeurIPS, 2023.
> > >
> > > ***
> > >
> > > Thank you for your time and attention to detail. We will incorporate all the additional results and discussions in our revision.

---

### Official Review · Reviewer_Jrjz · 2026-03-08

**Soundness:** 2
**Presentation:** 1
**Significance:** 2
**Originality:** 2
**Overall Recommendation:** 3
**Confidence:** 2

**Summary:**

The paper proposes T3CEN, a novel architecture designed to enhance the robustness of CNNs against color distribution shifts (e.g., color jittering, illumination changes). The authors identify that current color-equivariant networks, while handling Hue as a rotation, treat Saturation and Luminance as 1D translations, leading to boundary truncation artifacts in interval-valued data. To address this, the authors introduce a "double-cover mapping" that folds and wraps these linear intervals into a hypertoroidal space, theoretically achieving "perfect equivariance" across all HSL dimensions. The method is validated on datasets like Camelyon17, demonstrating potential for OOD generalization.

**Compliance With Llm Reviewing Policy:**

Affirmed.

**Final Justification:**

The reason I lean towards to negative attitude to this paper is that the Question 1 is still not fully resolved: the rebuttal clarifies how the lifted representation transforms by cyclic permutation, but it still does not explain why bounded saturation and luminance themselves should be viewed as physically or semantically cyclic variables in the original HSL space, so the double-cover still reads more as a mathematical device for exact equivariance than a faithful physical symmetry. In addition, the new ablation is useful, but it still does not cleanly resolve the capacity confound in the main multi-dimensional comparison.

**Key Questions For Authors:**

- Could the authors clarify the physical/semantic meaning of the lifted cyclic actions on bounded saturation and luminance spaces, and explain why equivariance under these actions is practically meaningful?

- Could the authors provide a capacity-controlled ablation that keeps channel width/filter depth fixed, to disentangle the effect of equivariance from reduced representation capacity?

**Limitations:**

yes

**Strengths And Weaknesses:**

Strength:
- The use of double-cover mapping to transform finite interval transformations into cyclic group actions on a torus offers a novel topological perspective on handling physically bounded quantities.
- The framework shows promising performance improvements in OOD scenarios where color shifts are prevalent, such as in histological image analysis.

Weaknesses:
1. **Notation and exposition are not sufficiently self-contained.** Several key quantities are introduced without immediate local clarification. In Section 4, symbols such as $N,M,R$, the space $\mathbb{T}^1$, and the function $f$ in Eq. (6) are used with limited on-the-spot explanation of their meaning or domain. The luminance construction is also deferred to Appendix A.1 rather than defined inline. This weakens readability and makes the derivation harder to follow.
2. **The physical interpretation of the double-cover construction remains under-justified.** The topological construction is mathematically elegant, but the paper does not clearly explain why the induced cyclic symmetry is semantically faithful to bounded saturation and luminance in HSL space. Figure 1 already shows that luminance lifting can map a white background to black or gray backgrounds, and Section 5.4 further notes input-dependent coverage and degenerate lifts. The method is therefore well motivated algebraically, but its physical interpretation is not fully resolved.
3. **The experimental control over model capacity is not clean enough.** In the Camelyon17 study, the authors keep parameter count fixed across HSL configurations by reducing filter depth as the equivariance order increases, and explicitly acknowledge that this can reduce capacity and make higher-order models perform worse. As a result, the degradation of T$^3$CEN-H4S4 relative to T$^3$CEN-S4 in Table 5 cannot be attributed solely to the added multi-dimensional equivariance constraints.
4. **The baseline set is relatively narrow for the claimed robustness and generalization scope.** The empirical study mainly compares against ResNet, CEConv, and LCER variants, while the related-work section discusses broader robustness strategies such as AugMix and DeepAugment. This makes it difficult to assess how competitive the proposed architecture is relative to a wider range of established robustness or domain-generalization approaches.

---

> ### Author Rebuttal · Authors · 2026-03-31
>
> Thank you for the time and valuable suggestions. We are encouraged that you found our math formulation to be sound and see the potential for OOD generalization. We appreciate your questions on the ituition of double cover and critiques of presentation and baselines, and hope to address them below. We will include all feedback in our revision.
>
> >W1. Notation and exposition.
>
> We agree that Sec. 4 can benefit from more local clarification. In particular, $N$, $M$, and $R$ denote the discretization orders of the hue, saturation, and luminance groups, respectively; $\mathbb{T}^1$ denotes the circle (1-torus); and $f^l$ denotes the feature map at layer $l$, as introduced in Equation (1). We will revise Section 4 so that these quantities are defined explicitly when used rather than rely on the reader to infer them from context.
>
> We also agree that deferring the luminance formulation to Appendix A.1 makes the section less self-contained. In the revision, we will move the luminance inline so that the hue, saturation, and luminance formulations are presented in parallel.
>
> >W2/Q1. Physical interpretation.
>
> The purpose of the double cover is to give interval-valued quantities a well-defined group structure so that group convolution can be applied without the clipping artifacts that arise when they are modeled as translations. In LCER, the saturation translation on features introduces zero padding (see Eq. 6), which leads to non-equivariant outputs (see Fig. 2).
>
> In our construction, saturation and luminance are lifted to a cyclic space, and the lifted representation transforms by exact cyclic permutation under the corresponding action (see Fig. 2). The practically meaningful property is therefore not that every individual lifted image corresponds to a canonical saturation/luminance shift, but that real saturation/luminance shifts of the input induce a predictable cyclic permutation of the lifted orbit, which is exactly the structure required by group convolution.
>
> We agree that the original Fig. 1 was not especially illustrative for this point. In particular, it emphasized individual lifted images rather than the orbit-level permutation structure. Furthermore, the luminance example used white background pixels, which are a degenerate edge case under luminance lifting (see Figure 5). This produced the white-to-gray transformations that you correctly pointed out.
>
> We include a clearer visualization in the anonymous link https://imgur.com/a/xc2C9Az, which shows the cyclic permutation of the lifted representation under input HSL shifts. We will replace Fig. 1 and revise the discussion to make clarify this distinction.
>
> >W3. Capacity-controlled ablation.
>
> To preserve per-channel expressivity at higher orders, T3CEN needs to be modified so that the number of filters per lifted representation remains fixed, rather than reduced to preserve overall parameter count. To address this point, we include an additional comparison below using a filter-depth-constant T3CEN-H4S4-unconstrained on CIFAR-10. We find that T3CEN-H4S4-unconstrained performs similarly to T3CEN-S4. These accuracies differ from those reported in Table 6 because these rebuttal models were only trained for 100 epochs due to the compute associated with the unconstrained T3CEN.
>
> ||T3CEN-S4|T3CEN-H4S4-constrained|T3CEN-H4S4-unconstrained|
> |-|:-:|:-:|:-:|
> |Err.|18.4|24.4|17.6|
> |it/s|3.7|1.7|0.8|
> |Param|2.6M|2.6M|10.5M|
>
> A more principled way to quantify this tradeoff would be to gradually increase the filter count of T3CEN-unconstrained until its error matches that of single-channel T3CEN. The required increase in parameter count would then quantify the additional expressivity needed to recover that performance. We were not able to complete this analysis during the rebuttal period because of the associated computational cost, but we agree that it would be a useful addition for the revision.
>
> >W4. The baseline...narrow.
>
> We agree that stronger augmentation baselines would strengthen the empirical evaluation and better position T3CEN relative to the broader literature on robustness. To address this point, we evaluate T3CEN against stronger augmentation baselines AugMix [1], DeepAugment [2], and Planckian Jitter [3]. Under saturation shift, the best T3CEN model outperforms AugMix/Planckian Jitter on 5 of 6 datasets and DeepAugment on all 6 datasets. Under luminance shift, the best T3CEN model outperforms all AugMix/DeepAugment on all 6 datasets. These results suggest that explicitly incorporating color geometry as a hard constraint can yield stronger accuracy and generalization than augmentation-based approaches. We include detailed results in our reply to reviewer qJ2k W1.
>
> We will include these result and add discussion positioning T3CEN more clearly relative to augmentation-based robustness methods. Thank you for encouraging us to strengthen the evaluation.
>
> [1] Dan Hendrycks, et al. ICLR, 2020. [2] Dan Hendrycks, et al. ICCV, 2021. [3] Simone Zini, et al. ICLR, 2023.

---

> > ### Author Rebuttal · Reviewer_Jrjz · 2026-04-03
> >
> > I thank the authors for the detailed rebuttal, the revised visualization, and the additional experiments. These clarifications partially address my concerns and lead me to raise my score from 2 to 3. However, my overall recommendation remains negative. Most importantly, Question 1 is still not fully resolved: the rebuttal clarifies how the lifted representation transforms by cyclic permutation, but it still does not explain why bounded saturation and luminance themselves should be viewed as physically or semantically cyclic variables in the original HSL space, so the double-cover still reads more as a mathematical device for exact equivariance than a faithful physical symmetry. In addition, the new ablation is useful, but it still does not cleanly resolve the capacity confound in the main multi-dimensional comparison.

---

> > > ### Author Response · Authors · 2026-04-05
> > >
> > > Thank you for the reply and the chance to further clarify our submission. We will address your questions on semantic meaning of the double cover and capacity-controlled ablation below.
> > >
> > > > New ablation is useful, but it still does not cleanly resolve the capacity confound in the main multi-dimensional comparison.
> > >
> > > We apologize in advance if we misinterpreted the exact experimental setting your question refers to. We believe that your central critique is regarding the changing backbone (ResNet50->ResNet44), dataset (Camelyon->CIFAR), and parameter count (23.5M->2.6M) of the reported ablation.
> > >
> > > To that end, we agree the rebuttal ablation does not fully resolve the Table 5 capacity confound. To address this concern more directly, we further evaluate the unconstrained T3CEN-H4S4 on Camelyon17 using the same ResNet50 backbone used in Table 5. This keeps the experimental setting aligned with the main comparison in Table 5 while removing the filter-depth reduction used to maintain parameter count. Under this capacity-controlled setting, we find T3CEN-H4S4-Unconstrained outperforms T3CEN-S4 slightly.
> > >
> > > ||T3CEN-S4|T3CEN-H4S4|T3CEN-H4S4-Unconstrained|
> > > |-|:-:|:-:|:-:|
> > > |Error|12.11|20.95|11.63|
> > > |Param.|23.5M|23.5M|93.8M|
> > >
> > > We therefore agree with the reviewer that the original fixed-parameter comparison confounds equivariance order with representational capacity. We hope this cleaner capacity controlled comparison better separates the benefits of higher-order equivariance from the effect of reduced expressive capacity.
> > >
> > > Interestingly, the magnitude of improvement reflects the true HSL distribution of the dataset, where hue-shift is much less pronounced than saturation-shift. We quantify this by analyzing the distribution shifts between the slides from the training and validation hospital - see https://imgur.com/a/DDx2I4c. Concretely,
> > > - Average hue is shifted by 0.0103
> > > - Average saturation is shifted by 0.0599
> > >
> > > Thank you for encouraging us to be more precise with out ablation study. We will include these additional results to help future readers understand the trade-off in T3CEN.
> > >
> > > > Does not explain why bounded saturation and luminance themselves should be viewed as physically or semantically cyclic variables
> > >
> > > We agree that saturation and luminance are not intrinsically cyclic variables in the native HSL space; they are bounded interval-valued quantities, and this is exactly why translation-based modeling (such as LCER) introduces clipping artifacts and breaks exact equivariance. Rather, the double cover introduces an auxiliary lifted space on which the action is cyclic and compatible with exact group convolution.
> > >
> > > Because the projection $\pi$ is nonlinear, individual elements of the lifted orbit need not be representable as a single additive saturation/luminance shift in native HSL coordinates, even though they remain valid interval-valued images (i.e. sat/lum$\in[0, 1]$). However, when an input image undergoes a valid bounded saturation or luminance change, its lifted orbit transforms by a predictable cyclic permutation (see our revised image in the original rebuttal: https://imgur.com/a/xc2C9Az). This is the structure required by group convolution and is what allows us resolve the clipping artifacts in baselines such as LCER.
> > >
> > > We will revise the paper to make this distinction explicit and to better connect it to existing cover-based constructions in geometric deep learning [3,4], which we currently mention only briefly in related work. In these work, covers are used to construct a better behaved representation space where operations are well defined.
> > >
> > > [3] Niv Haim, et al. "Surface networks via general covers." ICCV, 2019.\
> > > [4] Haggai Maron, et al. "Convolutional Neural Networks on Surfaces via Seamless Toric Covers." ACM Trans. Graph., 2017.
> > >
> > > ***
> > >
> > > Thank you for your time and constructive critique of our submission. We will include all additional experiments and discussions in our revision.

---

### Official Review · Reviewer_skvb · 2026-03-13

**Soundness:** 3
**Presentation:** 4
**Significance:** 3
**Originality:** 3
**Overall Recommendation:** 5
**Confidence:** 3

**Summary:**

In this work, colour equivariance is implemented using a lifting operation to compactify hue, saturation, and luminance. Various experiments show the effectiveness of the proposed mechanism and further analyses show additional properties of the model such as scale equivariance.

**Compliance With Llm Reviewing Policy:**

Affirmed.

**Final Justification:**

The authors addressed my concerns and ran additional experiments that answered my questions.

**Key Questions For Authors:**

1) Can you add a table or plot that shows the computational cost (both space/memory and time) of this model w.r.t. the others? Such a comparison would improve the quality of the paper significantly.
2) Is (partial) scale equivariance a fixed property of this model? If so, when and how can this cause issues when scale equivariance is not desired by the modeller? If not, can you briefly discuss why this model is expected to be able to have such ability? (I.e., is this effect expected/predicted by theory?) How does the scale equivariance error compare to other scale-equivariant models?
3) Are there tasks where colour equivariance might hurt performance? Can this penalty be quantified/estimated and compared with the drop in performance the models that are not (or imperfectly) colour equivariant undergo in the experiments conducted in this work?

**Limitations:**

yes

**Strengths And Weaknesses:**

Strengths:
- The presentation is very clear: Nice figures and tables, clean mathematical/theoretical exposition, and nice formatting. All of this made the paper a joy to read and easy to follow. Bravo.
- The experimental settings are varied and the results are promising (if not perfect) when compared to alternatives.
- The theoretical framework is apt and well-suited for this task.

Weaknesses:
- The paper is lacking an exploration of the computation cost of picking this model over others.
- Scale equivariance is briefly discussed but not placed in context with other models.

---

> ### Author Rebuttal · Authors · 2026-03-31
>
> Thank you for your time and thoughtful comments. We are encouraged that you found T3CEN to be theoretically apt and shows promising results on a variety of experiments. We also acknowledge your concerns and address these below, and will incorporate all feedback in the revision.
>
> >W1/Q1. computation cost
>
> We agree that a clearer evaluation of computate cost would help quantify the cost-performance tradeoff in group convolutions [1]. We include training time and memory usage below. We note that T3CEN/LCER-H1 reduces to ResNet.
>
> ||H1|H4|S4|L4|H4S4|H4L4|H4S4L4|
> |:-|:-:|:-:|:-:|:-:|:-:|:-:|:-:|
> |MiB-T3CEN|1030|2739|2739|2739|9480|9480|32842|
> |MiB-LCER|1014|2702|2702|2702|9906|9906|not possible|
> |iter/s-T3CEN|19.38|11.33|11.34|11.27|5.23|5.23|1.58|
> |iter/s-LCER|19.01|11.30|11.31|11.31|5.22|5.21|not possible|
>
> As with LCER, T3CEN introduces a similar level of additional compute/memory overhead. This increase becomes more pronounced for higher-order joint HSL lifting. The empirical evaluation in Table 6 indicates that T3CEN uses the additional compute more effectively compared to LCER. We will add this comparison to the revision to make the cost-performance tradeoff more explicit.
>
> [1] Taco Cohen and Max Welling. ICML, 2016.
>
> >W2. Scale equivariance...context?
>
> We included the discussion of scale equivariance to illustrate that the proposed hypertoroidal double-cover can help resolve imperfect approximations beyond color geometry. We agree, however, that this discussion is currently too brief and not sufficiently contextualized relative to existing scale-equivariant models [2,3]. We will revise this section to better explain its role as an extension of the lifting idea, and to place it more clearly in the context of prior work on scale equivariance.
>
> [2] Ivan Sosnovik, et al. ICLR, 2020. [3] Md Ashiqur Rahman and Raymond A. Yeh. NeurIPS, 2023.
>
> >Q2. Is scale a property of T3CEN? Baseline comparison?
>
> Scale equivariance is not a property of the proposed T3CEN architecture. We included the discussion on scale only to show that the hypertoroidal double-cover can also be applied to groups beyond saturation and luminance. Therefore, scale equivariance is not an implicit/unavoidable behavior of T3CEN as used in our paper; it would arise only if the same lifting construction were explicitly applied to scale.
>
> For this reason, we did not include a comprehensive comparison with dedicated scale-equivariant architectures such as [2,3]. We agree, however, that this point should be clarified more explicitly. We will move the discussion on scale to the appendix to distinguish more clearly between the core color-equivariant model studied in the paper and the separate proof-of-concept on scale. We apologize for the confusion.
>
> >Q3. Are there tasks where colour equivariance might hurt performance?
>
> Thank you for raising this interesting question. Yes, there are cases where color equivariance hinders performance. We mainly envisage two failure modes:
> 1. When there is little or no color distribution shift between the training and testing dataset. In this case color equivariance provides little extra information, and the reduced filter depth can reduce expressivity (see Appendix C.1).
> 2. When absolute color is itself predictive. Consider fruit ripeness classification, where color acts as a major indicator. Generalization across color in this setting may blur class boundaries and reduce classification performance.
>
> (1) We can quantify failure mode 1 using the hue-shifted MNIST dataset (see Appendix D.5 for details) trained on ResNet-44 and T3CEN-H4 for 1000 iteration. The training and testing set both consist of a single hue. We gradually increase the difference between the training and testing color, measured in terms of degrees on the hue wheel. We see that when no distribution shift occurs, ResNet-44 outperforms T3CEN-H4. With increasing hue-shift, T3CEN-H4 eventually out-performs ResNet.
>
> |Shift|ResNet|T3CEN-H4|
> |-|:-:|:-:|
> |$0^{\circ}$|98.38|94.18|
> |$5^{\circ}$|97.94|95.31|
> |$10^{\circ}$|97.23|96.10|
> |$15^{\circ}$|97.06|97.75|
>
> We will include a more systematic and robust analysis based on KL divergence between train and test sets in our revision.
>
> (2) We can quantify failure mode 2 using the KUTomaData Dataset [4], designed to benchmark binary classification on tomato ripeness (ripe/unripe). In this setting, absolute color has a strong correlation with class label. We train ResNet-18 and T3CEN-H4 for 100 epochs, and find that T3CEN-H4 yields classification accuracy significantly lower than vanilla ResNet.
>
> ||ResNet|T3CEN-H4|
> |-|:-:|:-:|
> |Acc.|80.87|68.25|
>
> Thank you for encouraging us to stress test T3CEN to truly understand the limitations of its application. We will revise our draft to include these discussions in the limitations sections.
>
> [4] Asim Khan, et al. Scientific Reports, 2023.

---

> > ### Author Rebuttal · Reviewer_skvb · 2026-04-03
> >
> > I wish to sincerely thank the authors for addressing my concerns with their original manuscript and running additional experiments in order to support their answers to my questions. I am happy with their responses.
> >
> > Comment: The confusion regarding the scale equivariant property of the model is an interesting one. I believe it is due to the fact that you were testing color equivariance extensively -> used the toroidal representation -> found that it worked well for scale, rather than frame the paper as an application of Cohen (2016), i.e. toroidal equivariance -> applied to both color and scale.
> >
> > I have augmented my score accordingly.

---

> > > ### Author Response · Authors · 2026-04-04
> > >
> > > Thank you for your reply and the positive assessment of our rebuttal. We will revise the draft to include the above discussions on
> > > - computational cost
> > > - limitations of enforcing full color equivariance
> > > - context of scale equivariance
> > >
> > > We appreciate your comment. Indeed the detour on scale originated from our experimentation/analysis on color equivariance - specifically we wanted to see if the toroidal cover can be extended to non-color groups. Thank you for the suggestion on how to frame our exploration, we will clarify scale's context w.r.t. color equivariant T3CEN.
> > >
> > > We thank the reviewer for their time and attention to detail during the review. We will incorporate all questions and suggestions in our revision.

---

### Official Review · Reviewer_Ppz8 · 2026-03-13

**Soundness:** 3
**Presentation:** 3
**Significance:** 3
**Originality:** 3
**Overall Recommendation:** 4
**Confidence:** 2

**Summary:**

The paper proposes a neural network architecture to achieve perfect color equivariance under hue, saturation, and luminance transformations. The authors point out that earlier color-equivariant models often approximate the geometry of color space using planar symmetries, which can lead to distortions. To overcome this, they propose using a hypertoroidal covering map that represents the color manifold with a cyclic topology. They show that their approach achieves near-zero equivariance error and improves out-of-distribution robustness on datasets like 3D Shapes, Small NORB, and Camelyon17.

**Compliance With Llm Reviewing Policy:**

Affirmed.

**Final Justification:**

My remaining concerns were addressed during the rebuttal. So I maintain my positive score of 4.

**Key Questions For Authors:**

N/A

**Limitations:**

Yes

**Strengths And Weaknesses:**

**Strengths**
- The paper is clear and well-motivated. It is also easy to follow and understand
- Using a double-cover map to wrap bounded intervals (like saturation $[0,c]$) onto a continuous circle is a clean way to bypass the boundary clipping issues.
- The work fits naturally into the broader literature on equivariant neural networks. Extending symmetry considerations to color geometry is a meaningful direction and aligns with recent efforts to incorporate physical or geometric priors into model design.
- Empirically, this approach gives higher prediction accuracy in the presence of color shifts and imbalance.

**Weaknesses**
- While the paper argues that existing methods only approximate the correct color geometry, it is not clear that this geometric mismatch is a significant bottleneck in practice. Simple approaches such as color jitter augmentation often already handle distribution shifts effectively. The paper does not convincingly show that modeling the color manifold with a hypertoroidal covering produces substantial empirical gains relative to these simpler baselines.
- The double-cover map (e.g., $\pi(\theta) = c \sin(\theta/2)$) is a many-to-one mapping in reverse. The authors admit this leads to degenerate cases, drastically dropping the information entropy of the representation (Figure 5). Pushing redundant information through an already expensive group convolution wastes massive amounts of computational capacity.
- The formulation assumes that shifts in hue, saturation, and luminance operate as independent and additive group actions. In the physics of real-world image formation (e.g., the Lambertian reflectance models), illumination changes are fundamentally multiplicative and coupled.

---

> ### Author Rebuttal · Authors · 2026-03-31
>
> Thank you for your constructive feedback. We are motivated that you found T3CEN to be well motivated and performant, while extending equivariant ML in meaning directions. We also acknowledge your concerns, especially on augmentation and compute, and hope to address these below.
>
> >W1. Augmentation.
>
> We appreciate this concern and agree that augmentation can be effective; however, our results show that embedding the color geometry provides measurable benefit. In Table 6, we compare T3CEN to ResNet-aug (ResNet+sat/lum jitter) under sat/lum shift. T3CEN outperforms ResNet-aug on 11 of 12 datasets. We also note a typo in the reported result for Caltech-101 under luminance shift: ResNet-aug achieves 57.03% error, not 30.82%. We thank the reviewer for helping us catch this mistake. In the revision, we will rename ResNet-aug to ResNet-jitter for clarity.
>
> We evaluate T3CEN against stronger augmentation baselines of AugMix (AM) [1], DeepAugment (DA) [2], and Planckian Jitter (PJ) [3]. Under sat-shift, the best T3CEN outperforms AM/PJ on 5 of 6 datasets and DA on all 6 datasets. Under lum-shift, the best T3CEN outperforms AM and DA on all 6 datasets. These results demonstrate that explicitly modeling color geometry as hard constraints can yield greater accuracy and generalization than augmentation approaches.
>
> |Sat-Shift|Caltech|CIFAR-10|CIFAR-100|Cars|Pets|STL|
> |-|:-:|:-:|:-:|:-:|:-:|:-:|
> |ResNet-AM|35.53|11.80|46.93|30.21|44.84|23.57|
> |ResNet-DA|42.87|20.39|51.64|42.32|43.57|24.17|
> |ResNet-PJ|38.81|12.28|51.33|31.25|41.43|21.83|
> |Best T3CEN|40.02|11.59|44.27|29.83|38.91|21.00|
>
> |Lum-Shift|Caltech|CIFAR-10|CIFAR-100|Cars|Pets|STL|
> |-|:-:|:-:|:-:|:-:|:-:|:-:|
> |ResNet-AM|57.30|40.84|70.60|75.50|74.76|54.50|
> |ResNet-DA|52.90|47.86|69.76|75.30|76.27|46.58|
> |Best T3CEN|46.23|25.33|52.57|55.55|62.85|38.01|
>
> We will include these stronger baselines in our revision and evaluate PJ on luminance shift. Thank you for pushing us to strengthen our evaluation.
>
> [1] Dan Hendrycks, et al. ICLR, 2020. [2] Dan Hendrycks, et al. ICCV, 2021. [3] Simone Zini, et al. ICLR, 2023.
>
> >W2. Degenerate...compute cost.
>
> We agree that degeneracy should be analyzed carefully; however, we believe the impact is more limited. We would like to clarify that these degenerate lifted representations arise from the double-cover at the extremities of the sat/lum scale. A similar issue also exists in the naive method from LCER: clipping at the extremities also produces lifted representations with low entropy. We also note that the drop in entropy shown in Fig. 5 is for a single pixel, not the entire input image.
>
> We do not believe that these degenerate representations induce significant compute burden. As shown in Figure 5, fully degenerate representations occur only when the input pixel has sat/lum of 0.5, and partially degenerate representations occur at 0.25/0.75. In CIFAR-100, only 0.57% of input pixels yield fully degenerate representations, while 0.64% yield partially degenerate representations. Stats for datasets in Tab. 6 is availible in our rebuttal to W2 of reviewer qJ2k.
>
> Moreover, Figure 6 suggests a practical way to mitigate this: selecting the lifting order with the highest entropy based on the input HSL distribution. We will clarify this point in the revision and include quantitative measurements of degenerate representations for all datasets studied.
>
> To quantify the effects of degenerate representations on performance, we train T3CEN-S4 a preprocessed CIFAR-100 dataset. We apply a non-zero sat-jitter to input pixels with 0.25/0.5/0.75 to remove degenerate representations. We show that T3CEN performs similarly on the original and preprocessed dataset. We will include this comparison with variance in our revision.
>
> ||Original|Preprocessed|
> |-|:-:|:-:|
> |T3CEN-S4|44.3|44.6|
>
> >W3. Illumination...coupled modeling.
>
> We agree that, from a photometric standpoint [4], modeling real-world image formation (eg. Lambertian reflectance) would require multiplicative/coupled transformations, and such models have been studied in prior work [5]. However, this is subtly different from the setting we study. T3CEN is applied to images and is designed to be equivariant to image-level shifts in hue, saturation, and luminance, which are variables in colorimetry [4].
>
> Our goal is therefore not to model the full physics of image formation, but to improve robustness to color shifts in the observed images themselves. Empirically, we show that T3CEN improves classification under color shifts arising from staining-process variation (Camelyon17, Table 5), rendering-parameter differences (3D Shapes, Table 1), and illumination-strength changes (Small NORB, Table 3).
>
> We will expand the discussion to ground T3CEN more clearly in the broader literature on variations in perceptual information, including color and photometric variation.
>
> [4] Günther Wyszecki and Walter Stanley Stiles. John Wiley & Sons, 2000. [5] Mounir Messaoudi, et al. ICLR, 2026.

---

> > ### Author Rebuttal · Reviewer_Ppz8 · 2026-04-02
> >
> > I thank the authors for their clarifications and additional experiments. However, the reported results on CIFAR-10, Caltech, CIFAR-100, and other datasets are noticeably lower than what is typically reported in the literature. It would help to know which ResNet variant was used (in terms of depth and width) and to have a clearer description of the training setup, as this is currently hard to infer. Further, it is unclear if T3CEN would still provide gains when paired with stronger, well-tuned baseline models.

---

> > > ### Author Response · Authors · 2026-04-04
> > >
> > > Thank you for your comment and the opportunity for us to continue the discussion. We will address your further concerns below.
> > >
> > > >ResNet variant?
> > >
> > > Thank you for raising this concern. We include details on ResNet version for CIFAR/Caltech/Pets/Cars/STL in Tab. 8 in Appendix D. Specifically, CIFAR-10/100 use ResNet44, Caltech/Pets/Cars/STL-10 use ResNet18, and Camelyon use ResNet50. Our implementation follows exactly from Table 1 in [5], where
> > > - ResNet-18 has (2,2,2,2) convolutional blocks each with (64,128,256,512) width
> > > - ResNet-44 (for CIFAR [6]) has (7,7,7) convolutional blocks each with (32,64,128) width
> > > - ResNet-50 has (3,4,6,3) convolutional blocks with bottleneck defined in [5]
> > >
> > > T3CEN share the same backbone as ResNet18/44/50 in terms of depth/width/residual connections, except standard convolutions (i.e. conv2d) are replaced with our fully color equivariant group convolution (the standard implementation introduced in [7]). All ResNet/T3CEN models are trained from scratch. We will revise our draft to make the architecture description more explicit in the main body.
> > >
> > > [5] Kaiming He, et al. CVPR, 2016.
> > > [6] Sergey Zagoruyko and Nikos Komodakis. BMVC, 2016.
> > > [7] Taco Cohen and Max Welling. ICML, 2016.
> > >
> > > >Training setup.
> > >
> > > We include details on training parameters for CIFAR/Caltech/Pets/Cars/STL in Table 8 Appendix D and for 3D Shapes/Small NORB/Hue Shift MNIST/Camelyon in Appendix D.1-D.6. We use the exact same training setting for both T3CEN and ResNet. In our revised draft we will include a consolidated table for all experiments reported in the paper, specifically
> > >
> > > ||CIFAR|Caltech/Pets/ Cars/STL|3D Shapes|NORB|MNIST|Camelyon|
> > > |-|:-:|:-:|:-:|:-:|:-:|:-:|
> > > |Epoch|300|300|10000 its|300|5|10000 its|
> > > |Batch Size|128|16|128|16|128|32|
> > > |LR|1E-1|1E-2|1E-4|1E-2|1E-3|1E-2|
> > > |Opt.|SGD|Adam|Adam|Adam|SGD|Adam|
> > > |Sch.|Cos. Anneal|n/a|n/a|n/a|n/a|n/a|
> > >
> > > >Reported...noticeably lower
> > >
> > > We agree that accurate baseline comparison is important to understand the true benefit of T3CEN. We were unsure which table your question was directed at, so we will clarify both Table 6 and Table 7 below.
> > >
> > > Table 6 report classification error on distribution-shifted dataset, where saturation/luminance is decreased by 0.5 (values below 0.0 are clipped to 0.0). Because the test distribution deliberately differs from training, the error rates are expected to be higher than those reported in standard literature (e.g., [8]). The purpose of this table is not to match published benchmarks, but to measure how each model degrades under color shift — and here T3CEN consistently outperforms baselines.
> > >
> > > Table 7 reports classification error on the original (unshifted) datasets - we note a difference in accuracy compared to papers like [8]. Both ResNet and T3CEN are trained with the same hyperparameters, without per-dataset tuning or learning rate schedulers (other than cosine annealing for CIFAR with SGD). This is by design: our goal is a controlled comparison where the only variable is the convolution operator. This setup isolates whether perfect color equivariance alone drives improved performance, rather than conflating gains with training recipe differences. Consistent with the baselines in [9], we did not fine-tune the training for this reason.
> > >
> > > [8] Simon Kornblith, et al. CVPR, 2019. [9] Attila Lengyel, et al. NeurIPS, 2023.
> > >
> > > >Stronger, well-tuned baselines.
> > >
> > > We address this by adding comparison to ResNet50 and Ray Tuned ResNet18.
> > >
> > > **Finetuning**: due to time/compute constraint associated with finetuning, we report ResNet18-Tuned on Caltech and Cars, which have the highest accuracy gap. We use Ray Tune [10] over learning rate, batch size, optimzer (SGD/Adam/AdamW), and scheduler (StepLR/CosineAnnealing/OneCycle/ReduceLROnPlateau). We find that increased efforts of fine tuning alone does not account for the difference in performance.
> > >
> > > |Saturation|Caltech|Cars|
> > > |-|:-:|:-:|
> > > |ResNet18|56.29|37.34|
> > > |ResNet18-Tuned|50.11|32.84|
> > > |T3CEN|40.02|29.83|
> > >
> > > **ResNet50**: For the Caltech/Pets/Cars/STL datasets, we further compare against ResNet50, which has over twice the parameter compared to ResNet18/T3CEN (23.7M vs 11.2M), trained under parameters defined in the first question. We find that increased expressivity alone does not account for the difference in performance.
> > >
> > > |Saturation|Caltech|Cars|Pets|STL|
> > > |-|:-:|:-:|:-:|:-:|
> > > |ResNet18|56.29|37.34|57.92|30.28|
> > > |ResNet50|51.38|33.38|52.88|27.67|
> > > |T3CEN|40.02|29.83|38.91|21.00|
> > >
> > > |Luminance|Caltech|Cars|Pets|STL|
> > > |-|:-:|:-:|:-:|:-:|
> > > |ResNet18|73.91|83.68|80.61|56.92|
> > > |ResNet50|71.16|78.68|76.23|52.61|
> > > |T3CEN|46.23|55.55|62.85|38.01|
> > >
> > > These additional baselines indicate that explicitly constraining the network to respect color geometry significantly improves generalization compared network fine tuning and additional expressivity.
> > >
> > > [10] Richard Liaw, et al. arXiv, 2018.
> > >
> > > We thank the reviewer for their time and engagement during the rebuttal. We will incorporate all questions and suggestions in our revision.

---

### Decision · Program_Chairs · 2026-04-30

**Decision:**

Accept (regular)

**Comment:**

Overall, the reviewers are mainly positive about this paper. While there are some concerns regarding how relatable the theory is in practice, specifically, "saturation and luminance are not intrinsically cyclic variables in the native HSL space." In general, most reviewers are okay with such a gap in the theory and practice. The AC also thinks the authors should clarify when such a gap exists, i.e., the theory is mainly a motivation/inspiration for a useful model, as demonstrated in the good empirical performance. The authors should also do a better job in discussing the computation/memory overhead. Finally, as the work is "truly equivariant", the paper could also discuss existing works that have also achieved truly equivariant results for other groups. Overall, the AC thinks the contribution of this work outweighs the weaknesses and recommends acceptance; the authors should incorporate their response in the rebuttal into the paper.